# Lua-LLM: Learning Unstructured-Sparsity Allocation for Large Language Models

**Mingge Lu     Jingwei Sun**[*]  **Junqing Lin     Zechun Zhou     Guangzhong Sun**[*]
University of Science and Technology of China
mingge@mail.ustc.edu.cn, sunjw@ustc.edu.cn, linjunqing@mail.ustc.edu.cn,
zhouzechun@mail.ustc.edu.cn, gzsun@ustc.edu.cn

## Abstract

Large Language Models (LLMs) have demonstrated remarkable capabilities, yet their extensive parameter scales pose significant challenges for practical deployment. Unstructured pruning has emerged as an effective model compression strategy with minimal performance loss, which introduces fine-grained sparsity for weight parameters. While existing methods employ a layer-wise pruning strategy to avoid the complexity of global pruning for billion-scale LLMs, they require appropriate sparsity allocation for the layer-wise pruning objectives and often lead to suboptimal solutions for the overall model. In this paper, we propose Lua-LLM (**L**earning **u**nstructured-sparsity **a**llocation in LLMs), a learning-based global pruning framework that explores the optimal unstructured sparsity allocation. Unlike existing pruning methods, which primarily focus on allocating per-layer sparsity, Lua-LLM achieves flexible allocation for both layer-wise and intra-layer sparsity. Furthermore, Lua-LLM leverages a soft Top-K operator to approximate the importance-based mask selection mechanism, enabling efficient binary mask learning. Experimental results on LLaMA and OPT families demonstrate significant performance improvements over existing methods.

## 1   Introduction

Large Language Models (LLMs) [1, 25, 67, 79] have demonstrated remarkable performance across a wide range of downstream tasks in natural language processing [9, 68, 69]. However, their ever-increasing parameter scales require substantial memory and computational resources, posing major challenges for their practical deployment on various platforms and applications. For instance, LLaMA-3.1-405B model [24] requires more than 754 GB memory to store its parameters in half-precision (FP16) format, far surpassing available memory on resource-constrained devices. To make LLMs more accessible and efficient, considerable efforts have been made to compress these models, including pruning [3, 19, 33, 48, 64], quantization [20, 31, 73], and knowledge distillation [36, 56, 65]. Pruning is an effective model compression approach and has been applied successfully in various model structures [10, 32, 45, 60].

Unstructured pruning [18, 26, 50] selectively removes less critical weight parameters at the element granularity. This process introduces element-wise sparsity in weight matrices while maintaining minimal degradation in model performance. Conventional pruning methods [49, 61] propose a global pruning strategy, which solves an optimization problem to minimize the overall model loss. However, given the massive parameter scale in LLMs, these methods become impractical due to the substantial computational overhead. To address this, recent studies like SparseGPT [19] and Wanda [64] split the global pruning objective into multiple local subproblems, each of which focuses on minimizing layer-wise pruning error that can be solved faster. Despite their efficiency gains, these methods

---

[*]Corresponding authors

39th Conference on Neural Information Processing Systems (NeurIPS 2025).

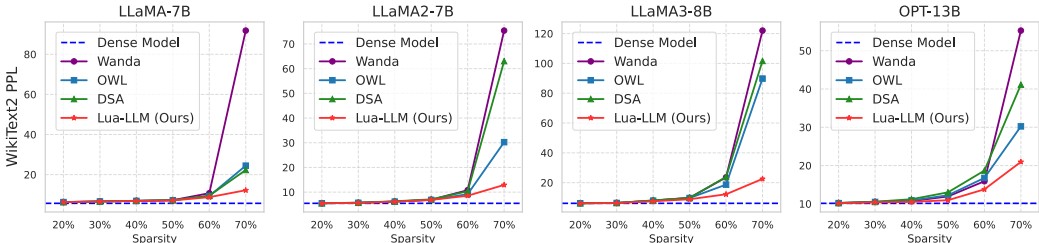

Figure 1: Model perplexity ↓ results on WikiText2 dataset with 2048 sequence length.

employ uniform pruning ratios across all layers and only focus on minimizing the local errors, posing the risk of removing important weights for more sensitive layers and leading to suboptimal solutions. Especially at high sparsity levels, these methods suffer severe performance degradation, undermining their practical applicability.

Recent studies like OWL [76] have thus focused on layer-adaptive sparsity allocation [29, 37, 38, 47, 76] to fulfill the potential of LLM pruning. While these techniques have demonstrated noticeable performance improvements over uniform pruning strategies, two substantial challenges remain: **First, existing layer-adaptive approaches overlook intra-layer sparsity allocation.** Following the pruning strategy in prior work [19, 64], these methods divide the weight matrix of a layer into finer-grained comparison groups, each employing a uniform sparsity level. Although BESA [74] aims to learn sparsity for these fine-grained groups, the substantial computational overhead caused by neural architecture search forces it to minimize the error of each transformer block, which still yields suboptimal solutions. **Second, layer-wise importance statistics are insufficient to indicate the inherent distribution of unstructured sparsity.** OWL uses layer outlier ratio as a heuristic proxy for per-layer sparsity allocations, which requires empirical sparsity mapping and lacks a solid theoretical foundation. DSA [37] conducts an evolutionary algorithm to find the optimal allocation function based on importance scores, yet the search process takes at least 12 hours for LLaMA-7B model, and the searched allocation function faces generalization issues for other models.

To address these challenges, we propose **Lua-LLM**, a gradient-based global pruning framework that learns fine-grained sparsity allocation to minimize the overall model performance loss. Our key insight is that Wanda's pruning strategy primarily focuses on activation sparsity in the input dimension, while its uniform row-wise sparsity configuration overlooks output sparsity, leading to an imbalanced sparsity distribution. Thus, we develop our Lua-LLM method by building upon an **adaptive row-wise sparsity allocation problem**. Lua-LLM decomposes the selection of overall pruning masks into row-wise pruning subproblems, each of which leverages a Top-K selection operator to **represent the mask selection process based on existing element-wise importance metric**. The non-differentiable Top-K operator is approximated using Sigmoid function, where the row-wise sparsity is transformed into a single pruning threshold parameter, enabling parameter-efficient mask learning. After optimizing the row-wise threshold parameters with end-to-end model performance loss, Lua-LLM identifies a sub-network within the original model, which achieves adaptive sparsity allocation for both layer-wise and intra-layer sparsity. For LLaMA-7B model, Lua-LLM learns sparsity allocation in only 1 hour on 2× NVIDIA A100 GPUs, which demonstrates superior efficiency compared to DSA.

We evaluate Lua-LLM on several LLMs, including LLaMA-7B/13B, LLaMA2-7B/13B, LLaMA3-8B, and OPT-6.7B/13B. Compared to existing sparsity allocation methods, Lua-LLM achieves significant performance improvements for compressed models, particularly at high sparsity ratios. As shown in Figure 1, Lua-LLM reduces the perplexity on LLaMA3-8B under a 70% pruning ratio by 99.5, 67.29, and 79.14 compared to Wanda, OWL, and DSA, respectively. Under a 60% pruning ratio, Lua-LLM improves the average accuracy on LLaMA-7B by 4.92%, 3.47%, 4.09%, compared to Wanda, OWL, and DSA, respectively, while incurring only a 3.18% accuracy degradation from the original model. When integrated with SpInfer [16], a GPU inference framework for sparse LLMs, Lua-LLM achieves end-to-end inference speedup for the compressed models with 50% - 70% sparsity levels on an NVIDIA A100 80 GB GPU, ranging from 1.18× to 1.73×. Our experiment results demonstrate that Lua-LLM achieves more adaptive sparsity allocation, enhancing the practical applicability for LLM pruning at high sparsity levels.

## 2 Related Work

**Unstructured Pruning for LLMs.** Pruning has a long history [27, 34] and has been successfully applied to compress neural networks of various structures [10, 28, 32, 45, 60]. Compared to structured [2, 6, 21, 48, 55, 72, 78, 81, 84] and semi-structured [17, 80, 82] pruning strategies, unstructured pruning [4, 19, 52, 64, 83] introduces finer-grained sparsity and leads to minimal performance loss. SparseGPT [19] proposes a post-training pruning framework that computes Hessian metrics for weight elimination and update. Wanda [64] designs a novel pruning metric and pattern, which outperforms SparseGPT at 50% sparsity without any weight updates. SparseLLM [4] leverages auxiliary variables for the decomposition of the global pruning problem and achieves alternating optimization into the subproblems with global optimality. To achieve practical inference speedup for unstructured sparse neural networks on GPUs, multiple works [16, 22, 41, 71] have been proposed to optimize the kernel latency of sparse matrix-matrix multiplication (SpMM) operation.

**Adaptive Layer-wise Sparsity.** To address the limitation of prior uniform pruning methods [4, 19, 52, 64], recent studies have thus focused on layer-adaptive sparsity allocation techniques. Based on observation of the correlation between activation outliers and performance of LLMs, OWL [76] adjusts the per-layer sparsity ratio according to layerwise outlier distribution. ALS [38] estimates inter-layer correlations using information orthogonality and then employs linear optimization to selectively prune features in intermediate layers. DSA [37] conducts an evolutionary algorithm to find an allocation function that establishes a mapping from element-wise scores to sparsity ratios and generalizes across different models. AlphaPruning [47] uses heavy-tailed self regularization to allocate layerwise sparsity in a theoretical manner. ATP [29] reduces the process of determining sparsity rates for multiple layers to the determination of a single common difference hyperparameter with a monotonically increasing arithmetic progression.

**NAS-based Pruning for LLMs.** Neural Architecture Search (NAS) [15, 39, 43, 51, 58, 66] is a pivotal technique in machine learning that automates the design of optimal neural network architectures. Recent studies have applied NAS to automatically identify the optimal sparsity pattern in pre-trained LLMs. Using the evolutionary algorithm, Pruner-Zero [14] evolves symbolic pruning metrics. DSA [37] searches for effective sparsity allocation functions. Search-LLM [63] identifies structured sub-networks using mask mutation. The gradient-based method also serves as an important NAS strategy. DISP-LLM [23] formulates dimension-independent structured pruning as an optimization problem and uses Straight-Through gradient estimator [44] to enable mask learning. MaskLLM [17] incorporates Gumbel Softmax for differentiable sampling of semi-structured pruning masks. BESA [74] parameterizes local sparsity objectives with learnable combinations of candidate pruning rates and minimizes block-wise reconstruction error. Given the massive parameter scale of LLMs, searching for element-wise unstructured pruning masks is much more challenging due to prohibitively high computational costs.

## 3 Preliminary

### 3.1 Problem Formulation

Given a pre-trained large language model with parameter $\mathbf{W}$, pruning is formulated as a constrained optimization problem, which utilizes inputs $\mathbf{X}$ to derive a sparse model with a binary pruning mask $\mathbf{M}$ and possibly updated weights $\widehat{\mathbf{W}}$ that minimizes the task performance loss:

$$\min_{\mathbf{M}, \widehat{\mathbf{W}}} \mathcal{L}(\mathbf{X}; \mathbf{M} \odot \widehat{\mathbf{W}}) \quad \text{s.t.} \quad \text{Sparsity}(\mathbf{M}) = p, \tag{1}$$

where the pruning mask $\mathbf{M}$ is constrained with target sparsity $p$. Since jointly optimizing both the pruning mask and the remaining weights is an NP-hard problem [8], a popular approach is to separate the pruning problem into mask selection and weight reconstruction.

**Challenges.** The massive parameter scale of LLMs introduces significant computational overhead for global pruning optimization problems. To reduce the complexity of the global pruning problem, prior methods [4, 19, 52, 64] split the full-model pruning problem into layer-wise subproblems. Despite the efficiency gains, the layer-wise pruning strategy presents two primary limitations. Firstly, it focuses on minimizing the local pruning error of each linear layer, while the non-linear operations

in LLMs suggest that such a layer-wise approach may yield a suboptimal solution for the entire network [40]. Secondly, it requires handcrafting an appropriate sparsity ratio $p_\ell$ for each layer, as the individual contributions of layers to the final model performance exhibit significant variation [23, 48, 76]. In this paper, we aim to address these challenges with an end-to-end learning method, which achieves adaptive sparsity allocation while minimizing the overall performance loss.

### 3.2 Revisit Wanda Pruning

Wanda [64] is a one-shot layer-wise pruning method, which prunes model weights without the weight reconstruction process. However, it achieves superior model performance at 50% sparsity level compared to methods that require weight update, such as SparseGPT [19]. The success of Wanda underscores the effectiveness of a better mask selection strategy, including careful design of the weight importance metric and the comparison group.

First, to capture the importance of model weight, Wanda proposes a novel pruning metric. Formally, given weight matrix $\mathbf{W} \in \mathbb{R}^{C_{out} \times C_{in}}$ and its input activation $\mathbf{X} \in \mathbb{R}^{L \times C_{in}}$, where $L$ is the input sequence length and $C_{out}$, $C_{in}$ are the output and input dimension, respectively, the importance score for weight element $\mathbf{W}_{ij}$ is computed as below:

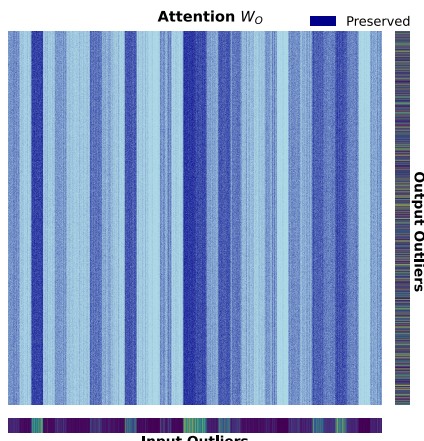

**Attention** $W_O$ — Preserved

**Output Outliers**

**Input Outliers**

$$\mathbf{S}_{ij} = |\mathbf{W}_{ij}| \cdot ||\mathbf{X}_{:,j}||_2, \tag{2}$$

where $|\mathbf{W}_{ij}|$ is the weight magnitude and $||\mathbf{X}_{:,j}||_2$ is the $\ell_2$ norm of the $j$-th column of input feature.

Figure 2: Visualization of the weight matrix $W_O$ under Wanda sparsity pattern and the corresponding input and output magnitude outliers.

Second, Wanda chooses output channels, which refer to the rows in a weight matrix, as the groups for weight importance comparison and applies a uniform sparsity level to all rows. Through extensive experiments, Wanda demonstrated that using row-wise comparison groups outperforms alternative configurations, including the entire matrix, single columns, multiple columns, or multiple rows.

**Observations.** Prior works [13, 42, 76] have demonstrated that activations in LLMs exhibit heavy-tailed distributions, characterized by a subset of features with exceptionally large outliers. These outliers have been proven to play a vital role in the remarkable performance of LLMs. From Figure 2, we notice that Wanda's sparsity pattern maintains low sparsity in the input dimensions corresponding to these outliers, which provides an intrinsic explanation for its effectiveness. However, Wanda misses a potential issue that the output dimensions also contain outlier magnitudes, while using the same row-wise pruning ratio might eliminate important weights in the output dimensions that are extremely sensitive, particularly at high sparsity levels.

**Motivating Study.** To verify our hypothesis, we perform a preliminary study for the output sparsity in weight matrices. Formally, given weight matrix $\mathbf{W} \in \mathbb{R}^{C_{out} \times C_{in}}$ and its input $\mathbf{X} \in \mathbb{R}^{L \times C_{in}}$, the output $\mathbf{Y} \in \mathbb{R}^{L \times C_{out}}$ is computed as below:

$$\mathbf{Y} = \mathbf{X}\mathbf{W}^\top. \tag{3}$$

We collect the output magnitude $\mathbf{y} \in \mathbb{R}^{C_{out}}$, where its $j$-th element $\mathbf{y}_j = ||\mathbf{Y}_{:,j}||_2$ is the $\ell_2$ norm of the $j$-th column of output feature.

We first apply the Wanda pruning strategy to uniformly prune the rows with 80% sparsity level for weight matrices in LLaMA-2-7B model. Then we adjust the row-wise sparsity ratios for each layer as follows: according to the output magnitude $\mathbf{y}$, we reduce the sparsity to 60% for the top 128 rows, and increase the sparsity to 100% for the bottom 128 rows. Although completely removing some output channels will introduce structured sparsity, which empirically leads to lower performance than unstructured sparsity, we still observe that the perplexity on WikiText2 [53] dataset decreases from 2335 to 1532, indicating that our heuristic strategy enhances the model performance. This non-uniform importance distribution in the output dimension motivates our pruning approach towards an adaptive row-wise sparsity pattern.

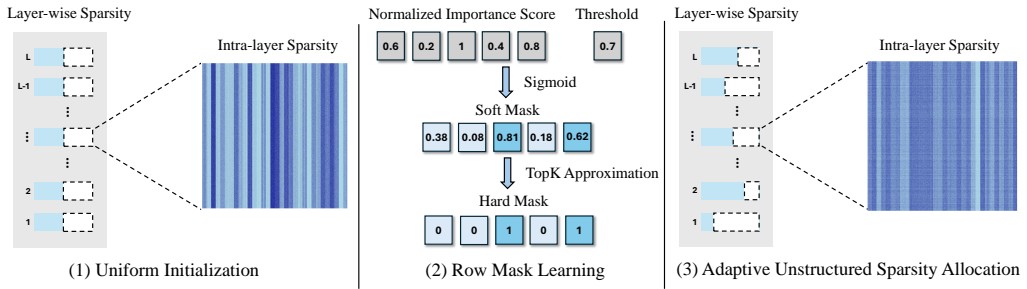

Figure 3: Overview of the proposed Lua-LLM pruning framework.

# 4 Lua-LLM: Learning Unstructured-Sparsity Allocation for LLMs

In this section, we motivate and describe our pruning method, Lua-LLM, which learns a more flexible and balanced sparsity allocation pattern from a global optimization perspective. Lua-LLM splits the global pruning mask selection problem into multiple row-wise subproblems, each of which is transformed into a single learnable threshold parameter using a soft Top-K operator, thereby enabling the end-to-end optimization of the model performance through gradient descent. An overview of our pruning framework is presented in Figure 3.

## 4.1 Row-Wise Mask Selection with Weight Importance Metric

To address the limitations of layer-wise pruning, we aim to identify optimal pruning masks by solving the global pruning optimization problem. However, directly learning the binary pruning masks for LLMs is prohibitively expensive. Motivated by our observation in Section 3.2, we reformulate the global pruning problem as an adaptive row-wise sparsity allocation problem. Within each row of weight matrices, we directly use the Wanda importance metric, which is introduced in Equation (2), and prune the less important weights according to the allocated sparsity. We first formulate the row-wise mask selection process as below.

Formally, let $\mathbf{w} \in \mathbb{R}^{C_{in}}$ denote a row in the weight matrix of a model layer, and let $\mathbf{s} \in \mathbb{R}^{C_{in}}$ represent the corresponding importance scores derived from the Wanda pruning metric. For a target sparsity ratio $p \in [0, 1]$, we define the pruning threshold $t \in \mathbb{R}$ as the $K$-th largest value in $\mathbf{s}$, where

$$K = \lfloor (1 - p) \cdot C_{in} \rfloor, \tag{4}$$

ensuring that exactly $K$ weights are retained. This threshold $t$ enforces the sparsity constraint:

$$p = \frac{1}{C_{in}} \sum_{i=1}^{C_{in}} \mathbb{I}(s_i < t), \tag{5}$$

where $\mathbb{I}(\cdot)$ is the indicator function. A mask selection function $f$, which retains the Top-$K$ most important weights, maps the importance score $\mathbf{s}$ to a binary mask $\mathbf{m} \in \{0, 1\}^{C_{in}}$ as follows:

$$m_i = f(s_i, t) = 0.5 \cdot \text{sign}(s_i - t) + 0.5 = \begin{cases} 1, & \text{if } s_i \geq t, \\ 0, & \text{otherwise}, \end{cases} \tag{6}$$

where $m_i = 1$ indicates that the corresponding weight $w_i$ will be preserved, and $m_i = 0$ indicates that the weight $w_i$ will be pruned.

## 4.2 Soft Approximation for Top-K Selection Function

After constructing pruning masks for each row of the weight matrices, we need to solve the adaptive row-wise sparsity allocation problem. However, the non-differentiable nature of the discontinuous mask selection function $f$ poses a major challenge for generating differentiable masks, preventing the use of common optimization solvers such as the gradient descent technique.

---

**Algorithm 1** The multi-stage Lua-LLM pruning algorithm.

---

**Input**: training dataset $\mathcal{X}$, pre-trained LLM model, and target sparsity $p$.

**Output**: unstructured sparse model.

1: **Initialization:** integrate mask modules into Attn and MLP layers, prepare uniform importance scores within $[0, 1]$, initialize all threshold parameters to target sparsity $p$.
2: **for** $t \leftarrow 1$ to $T$ **do**:
3:     **for** each weight matrix $\mathbf{W}^l$ **do**:
4:         Generate soft pruning mask $\mathcal{M}^l$ with row-wise thresholds $\{t_j\}_{j=1}^{C_{out}}$ by Eqn.(7),
5:     **end for**
6:     Forward propagation: $\mathcal{L}_{total} = \mathcal{L}_{task}(\mathcal{X}; \mathcal{M} \odot \mathbf{W}) + \lambda_{reg}\mathcal{L}_{reg}(\mathcal{M}; p)$,
7:     Update row-wise threshold parameters during back-propagation,
8: **end for**
9: Save the row-wise threshold parameters for each weight matrix,
10: **Pruning:** computing hard masks for pruning by Eqn.(6).

---

To address this challenge, we employ the straight-through-estimator (STE) technique [5], enabling gradient computation via a soft approximation when processing non-differentiable functions. Intuitively, we employ the sigmoid function to construct a soft approximation of the Top-K selection function, so that each row-wise pruning mask can be generated with a single learnable threshold parameter $t$. To mitigate the training instability caused by importance outliers (Section 3.2), we map the importance scores to a uniform distribution within $[0, 1]$. Then, given a row of weights $\mathbf{w} \in \mathbb{R}^{C_{in}}$ with the uniformly distributed importance scores $\mathbf{s} \in [0, 1]^{C_{in}}$, we reparameterize the target sparsity $p$ as a learnable threshold parameter $t$, and compute the soft pruning mask $\tilde{\mathbf{m}} \in [0, 1]^{C_{in}}$ with a differentiable mask selection function $\tilde{f}$:

$$\tilde{m}_i = \tilde{f}(s_i, t) = \sigma\big(\lambda(s_i - t)\big), \tag{7}$$

where $\sigma(x) = (1 + e^{-x})^{-1}$ is the sigmoid function and $\lambda$ controls the approximation level. We set the value of parameter $\lambda$ to $C_{in}$, which is large enough to provide a precise approximation. The soft alternative for the Top-K selection function makes the loss function $\mathcal{L}$ differentiable with respect to the pruning threshold $t$, and the global pruning problem can be optimized with a gradient descent method. Formally, the gradient for parameter $t$ can be computed as:

$$\frac{\partial \mathcal{L}}{\partial t} = \sum_{i=1}^{C_{in}} \frac{\partial \mathcal{L}}{\partial \tilde{m}_i} \frac{\partial \tilde{m}_i}{\partial t}, \quad \frac{\partial \tilde{m}_i}{\partial t} = \frac{\partial \sigma\big(\lambda(s_i - t)\big)}{\partial t}. \tag{8}$$

**Remark.** Equation (7) provides a row-wise differentiable mask constructed from the underlying pruning threshold $t$ and importance score $s$. Since the operation patterns across all rows are identical, we can leverage the broadcast computation mechanism in PyTorch to compute row-wise pruning masks in parallel for each weight matrix. Another gradient-based unstructured pruning method for LLMs, BESA [74], requires a customized CUDA operator to support the parallel generation for intra-layer probabilistic pruning masks, while Lua-LLM does not induce any backend modification.

## 4.3 Learning Row-wise Sparsity Allocation

With the proposed soft mask selection technique, we can formulate the global pruning optimization problem based on row-wise sparsity allocation and conduct the end-to-end learning process with the gradient descent method. To reduce our search cost and improve convergence, we employ a uniform initialization strategy, and then enforce the target sparsity with a regularization term.

**Initialization.** Before training, we first employ Wanda pruning method to obtain the importance scores for weight parameters. As introduced in Section 4.2, we map the importance scores to a uniform distribution within $[0, 1]$ to mitigate the training instability caused by importance outliers. With this preprocessing approach, we can directly initialize the row-wise threshold parameter $t$ as the target sparsity $p$, leading to a uniform sparsity initialization pattern.

**Learning Objective.** For a pre-trained large language model, we integrate learnable mask modules into the Attention and MLP layers while freezing the original model parameters, thus facilitating

Table 1: Model Perplexity ↓ results of different unstructured sparsity allocation methods evaluated on WikiText2 dataset with 2048 sequence length.

| Method | Sparsity | WikiText2 Perplexity ↓ | | | | | | |
|---|---|---|---|---|---|---|---|---|
| | | LLaMA-7B | LLaMA-13B | LLaMA2-7B | LLaMA2-13B | LLaMA3-8B | OPT-6.7B | OPT-13B |
| Dense | 0% | 5.68 | 5.09 | 5.47 | 4.88 | 6.14 | 10.86 | 10.13 |
| Wanda | | 7.25 | 6.15 | 6.92 | 5.97 | 9.82 | 11.98 | 11.93 |
| OWL | 50% | 7.22 | 6.08 | 6.86 | 5.92 | 9.68 | 12.21 | 12.23 |
| DSA | | 7.26 | 6.11 | 7.07 | 6.11 | 9.81 | 12.40 | 13.00 |
| Lua-LLM | | **7.12** | **6.05** | **6.85** | **5.89** | **8.87** | **11.96** | **10.94** |
| Wanda | | 10.73 | 8.77 | 10.79 | 8.38 | 23.58 | 15.22 | 15.90 |
| OWL | 60% | 9.35 | 7.67 | 9.18 | 7.56 | 18.68 | 15.54 | 16.77 |
| DSA | | 9.48 | 7.71 | 10.36 | 8.27 | 23.80 | 16.65 | 18.69 |
| Lua-LLM | | **8.75** | **6.97** | **8.54** | **6.89** | **12.21** | **14.46** | **13.79** |
| Wanda | | 91.83 | 56.26 | 75.42 | 45.63 | 122.02 | 157.48 | 55.26 |
| OWL | 70% | 24.46 | 16.95 | 30.23 | 20.57 | 89.81 | 42.92 | 30.23 |
| DSA | | 22.31 | 16.37 | 63.05 | 35.85 | 101.66 | 47.86 | 41.14 |
| Lua-LLM | | **12.21** | **9.26** | **12.92** | **8.98** | **22.52** | **37.45** | **20.96** |
| Wanda | | 2889.94 | 4008.78 | 2334.70 | 1134.94 | 894.99 | 4259.55 | 12516.03 |
| OWL | 80% | 1192.58 | 411.04 | 587.19 | 214.25 | 763.36 | 15370.94 | 5785.20 |
| DSA | | 934.29 | 316.81 | 1436.02 | 864.85 | 894.01 | 9664.09 | 533.89 |
| Lua-LLM | | **37.65** | **26.52** | **30.27** | **24.76** | **85.39** | **537.12** | **414.81** |

an end-to-end mask training process. To control the overall sparsity ratio, we introduce a sparsity regularization loss $\mathcal{L}_{reg}$ as below:

$$\mathcal{L}_{reg} = \begin{cases} \log\big(N(\mathcal{M})/(pN_{\text{total}})\big), & \text{if } N(\mathcal{M}) > pN_{\text{total}}, \\ 0, & \text{if } N(\mathcal{M}) = pN_{\text{total}}, \\ -\log\big(N(\mathcal{M})/(pN_{\text{total}})\big), & \text{if } N(\mathcal{M}) < pN_{\text{total}}, \end{cases} \tag{9}$$

where $N(\mathcal{M})$ is the number of removed weight parameters with pruning mask $\mathcal{M}$, $p$ is the target sparsity ratio, and $N_{\text{total}}$ is the number of weight parameters in the original model. The sparsity regularization loss encourages the sparsity of the overall model to converge to the target sparsity level $p$. The training objective is to minimize the language modeling loss $\mathcal{L}_{task}$ of next token prediction computed via the cross-entropy loss function. Combining the regularization loss $\mathcal{L}_{reg}$ with the language modeling loss $\mathcal{L}_{task}$, our total training loss in mask learning stage is:

$$\mathcal{L}_{total} = \mathcal{L}_{task}(\mathcal{X}; \mathcal{M} \odot \mathbf{W}) + \lambda_{reg}\mathcal{L}_{reg}(\mathcal{M}; p), \tag{10}$$

where $\mathcal{X}$ denotes the input tokens from training data, $\mathbf{W}$ represents the original weights, and $\lambda_{reg}$ is used to control the penalty for deviating from the target sparsity. Section 5.5 presents an ablation study on $\lambda_{reg}$, showing that our method achieves stable performance when $\lambda_{reg}$ is large enough.

## 5 Experiments

### 5.1 Experimental Settings

**Models.** We evaluate our Lua-LLM method on several LLMs as follows: LLaMA family [67]: LLaMA-7/13B, LLaMA-2-7/13B, LLaMA-3-8B; OPT family [79]: OPT-6.7B, OPT-13B.

**Datasets and Evaluation.** To train the learnable pruning masks, we use 2048-token segments from C4 [59] dataset, which is also used to sample calibration data in previous works. We evaluate the language modeling perplexity on the validation set of raw-WikiText2 [53] dataset. To ensure a fair comparison, the sequence length for all models is set to 2048. Following previous works, we also evaluate the zero-shot accuracy of pruned models on seven downstream tasks, including BoolQ [11], PIQA [7], HellaSwag [77], WinoGrande [62], ARC-easy [12], ARC-challenge [12], and OpenbookQA [54], based on the EleutherAI LM-Evaluation-Harness [35] framework.

**Baselines.** We compare our Lua-LLM method against Wanda [64], the uniform pruning method introduced in Section 3.2, and four adaptive sparsity allocation methods, OWL [76], DSA [37], AlphaPruning [47] and ATP [29].

Table 2: Zero-shot accuracy ↑ results on seven downstream tasks for the pruned LLaMA-7B, LLaMA2-7B and LLaMA3-8B models at 70% sparsity level.

| Model | Method | BoolQ ↑ | PIQA ↑ | HellaSwag ↑ | WinoGrande ↑ | ARC-e ↑ | ARC-c ↑ | OBQA ↑ | Mean ↑ |
|---|---|---|---|---|---|---|---|---|---|
| LLaMA-7B | Dense | 73.12 | 78.67 | 56.41 | 67.09 | 67.30 | 38.31 | 28.20 | 58.44 |
| | Wanda | 61.83 | 57.62 | 28.49 | 50.83 | 31.06 | 19.03 | 12.80 | 37.38 |
| | OWL | 62.97 | 64.53 | 34.78 | 56.67 | 42.30 | 24.74 | 16.40 | 43.20 |
| | DSA | 62.45 | 63.44 | 34.59 | 55.80 | 42.09 | 24.83 | 16.60 | 42.83 |
| | ATP | **65.79** | 67.46 | 37.44 | **61.43** | 50.63 | 25.68 | **20.80** | 47.03 |
| | Lua-LLM | 63.74 | **69.15** | **42.02** | 58.62 | **56.79** | **27.22** | 20.20 | **48.25** |
| LLaMA2-7B | Dense | 71.13 | 78.07 | 56.69 | 67.17 | 69.28 | 39.93 | 31.60 | 59.12 |
| | Wanda | 49.14 | 55.39 | 27.99 | 49.49 | 30.77 | 18.17 | 11.80 | 34.68 |
| | OWL | 62.23 | 62.19 | 31.88 | 55.41 | 43.77 | 20.39 | 16.80 | 41.81 |
| | DSA | 58.81 | 57.40 | 28.48 | 49.80 | 32.28 | 17.58 | 12.60 | 36.71 |
| | ATP | 62.39 | 66.81 | 36.08 | **61.01** | 50.76 | 23.38 | 20.40 | 45.83 |
| | Lua-LLM | **66.12** | **68.88** | **42.69** | 58.96 | **58.50** | **26.79** | **22.00** | **49.13** |
| LLaMA3-8B | Dense | 81.25 | 79.71 | 60.18 | 72.69 | 80.09 | 50.51 | 34.80 | 65.60 |
| | Wanda | 55.35 | 56.09 | 27.38 | 47.20 | 32.07 | 17.66 | 12.60 | 35.48 |
| | OWL | 61.90 | 58.22 | 28.37 | 50.43 | 35.35 | 17.15 | 13.20 | 37.80 |
| | DSA | 61.44 | 55.88 | 31.90 | 48.30 | 33.71 | 17.15 | 12.60 | 37.28 |
| | ATP | 61.79 | 62.18 | 31.46 | 54.93 | 41.79 | 20.39 | 16.40 | 41.28 |
| | Lua-LLM | **65.23** | **65.83** | **37.29** | **57.06** | **50.08** | **23.38** | **17.20** | **45.15** |

Table 3: WikiText2 perplexity ↓ results compared to AlphaPruning and ATP.

| Method | LLaMA-7B | | LLaMA2-7B | |
|---|---|---|---|---|
| | 70% | 80% | 70% | 80% |
| AlphaPruning | 23.86 | 698.56 | 28.87 | 1672.49 |
| ATP | 20.16 | 176.80 | 22.16 | 425.12 |
| Lua-LLM | **12.21** | **37.65** | **12.92** | **30.27** |

Table 4: End-to-end inference throughput (token/s) and speedup of sparse models.

| Model | Sparsity | Dense | 50% | 60% | 70% |
|---|---|---|---|---|---|
| OPT-6.7B | Throughput ↑ | 696.6 | 842.8 | 1012.7 | 1202.3 |
| | Speedup ↑ | - | 1.22× | 1.45× | 1.73× |
| OPT-13B | Throughput ↑ | 401.6 | 472.8 | 563.4 | 685.1 |
| | Speedup ↑ | - | 1.18× | 1.40× | 1.71× |

**Implementation.** We implement Lua-LLM in PyTorch [57] and use HuggingFace transformers library [70] for the evaluated LLMs. We integrate learnable mask modules into the Attention and MLP layers while freezing the original model parameters, thus facilitating an end-to-end mask learning process. We utilize the AdamW [46] optimizer with the learning rate set to $5 \times 10^{-3}$ and weight decay set to 0.05. The learnable threshold parameters are trained for 500 iterations, conducted on NVIDIA A100 80 GB GPUs.

## 5.2 Model Performance

In Table 1, we report the language modeling perplexity of pruned LLaMA and OPT models from 50% to 80% sparsity levels. We also compare our method to AlphaPruning and ATP in Table 3. Our Lua-LLM consistently outperforms the uniform Wanda pruning baseline and the state-of-the-art sparsity allocation methods. For example, under a 70% pruning ratio, Lua-LLM reduces the perplexity on LLaMA3-8B by 99.5, 67.29, and 79.14 compared to Wanda, OWL, and DSA, respectively. The performance improvement is particularly larger at higher sparsity levels. At the 80% sparsity level, Lua-LLM reduces the perplexity on LLaMA2-7B by 1642.22 and 394.85 compared to AlphaPruning and ATP, respectively. Notably, Lua-LLM achieves a perplexity result of 30.27 for the LLaMA-2-7B model at the 80% sparsity level, which is comparable to the performance of other methods at a lower 70% sparsity. Our experimental results demonstrate that directly extracting an unstructured sub-network from the original LLM can maintain acceptable performance with minimal degradation, even at higher sparsity levels.

In addition to model perplexity results, we report the zero-shot accuracy of LLaMA models at 70% sparsity level in Table 2. More results for 60% and 80% sparsity levels are shown in Table 11 and Table 12 of Appendix C.1, respectively. Under a 60% pruning ratio, Lua-LLM improves the average accuracy on LLaMA-7B by 4.92%, 3.47%, 4.09%, compared to Wanda, OWL, and DSA, respectively, while incurring only a 3.18% accuracy degradation from the original model. Our method achieves superior performance than baselines under various models, tasks, and sparsity levels, verifying our effectiveness in preserving important weights with adaptive sparsity allocation.

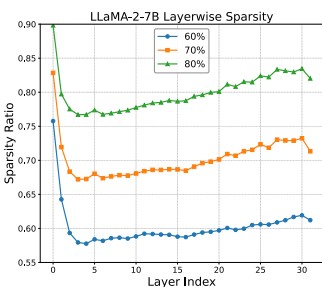
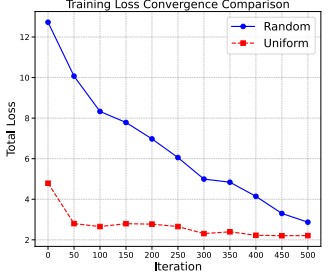
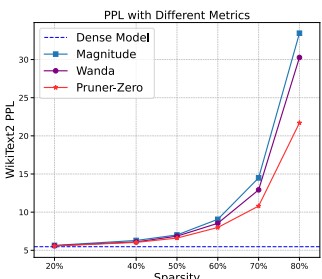

Figure 4: Layer-wise sparsity distribution for LLaMA2-7B.

Figure 5: Convergence for different initialization strategies.

Figure 6: WikiText2 PPL under different importance metrics.



Figure 7: Intra-layer sparsity allocation for Attention layers in the 70% sparse LLaMA2-7B model.

## 5.3 Inference Speedup

Unstructured pruning achieves high sparsity levels while maintaining minimal performance degradation, but the element-wise sparsity pattern it introduces cannot achieve a direct inference speedup on GPUs. Fortunately, a recent study proposes SpInfer [16], a sparse LLM inference framework specifically designed to optimize the kernel latency of sparse matrix-matrix multiplication (SpMM) operations at moderate sparsity levels on GPU Tensor Core architectures. We evaluate the pruned OPT-6.7B and OPT-13B models on an NVIDIA A100 80GB GPU and report the end-to-end inference throughput across different sparsity levels in Table 9. We use the testcase with batch size set to 8, input sequence length set to 32, and output sequence length set to 256. The compressed models achieve practical speedups on the GPU at sparsity levels from 50% to 70%, ranging from $1.18\times$ to $1.73\times$ compared to the dense model.

## 5.4 Sparsity Allocation

In Figure 4, we visualize the layer-wise average sparsity distribution of the pruned LLaMA2-7B model under varying global sparsity levels. Our results show that the distribution maintains a consistent pattern across different sparsity ratios, like prior studies [29, 37, 76], with layer-wise sparsity generally increasing across layers. Notably, we identify a distinct pattern that the first layer demonstrates substantially higher redundancy, and then the layer-wise sparsity exhibits a sharp decline within subsequent layers. The potential reason is that the sparsity levels of MLP modules in the first few layers are significantly higher than the average level. Additionally, we observe that the final layer exhibits a decreasing trend, deviating from the overall pattern.

We also show the intra-layer sparsity pattern in Figure 7, 10. Our method achieves adaptive intra-layer sparsity allocation for different types of modules, which provides an intuitive explanation for the significant performance improvement over prior methods. From Figure 7, we can observe the fine-grained channel-wise sparsity pattern within each self-attention head, which is compatible with findings in prior literature [30, 75]. Results for the layer-wise distribution of other models and the intra-layer sparsity pattern for MLP modules are shown in Appendix A. We hope that these observations can help researchers to gain insights into the inherent sparsity pattern of LLMs and design more adaptive one-shot model compression strategies in future work.

## 5.5 Ablation Study

**Search Efficiency.** We demonstrate the effectiveness of the initialization stage (Section 4.3) in helping Lua-LLM obtain the prior sparsity pattern of Wanda. We build a baseline that, instead of uniformly initializing the threshold parameters with the target pruning ratio, adopts randomly initialized thresholds. Figure 5 shows the convergence efficiency of the training loss within 500 iterations. We observe that training with uniformly initialized parameters significantly helps to reduce the training loss at the beginning, which proves the effectiveness of our initialization strategy. We further report the training costs of our Lua-LLM method in Table 5.

**Impact of Regularization Hyperparameter $\lambda_{reg}$.** As discussed in Section 4.3, the regularization loss hyperparameter $\lambda_{reg}$ is used to control the penalty for deviating from the target sparsity. In Table 6, we show the perplexity at 70% sparsity level with various hyperparameter values for LLaMA2-7B model. The results show that the performance of the learned sparsity pattern is stable when the hyperparameter $\lambda_{reg}$ is large enough. Specifically, if the regularization hyperparameter is not large enough, the regularization loss fails to converge to zero. This indicates that the pruning mask learned by our method does not meet the target sparsity, and the overall sparsity level is actually lower than the target one, since this can lead to better model performance and lower training loss. We mark the case as Not Converge, since it leads to an unfair comparison.

**Importance Metric.** We further explore whether an appropriate importance metric can enhance the performance of our methods. Specifically, we integrate Lua-LLM with three pruning metrics to prune LLaMA2-7B model, including Magnitude, Wanda[64], and Pruner-Zero [14]. In Figure 6, we compare the perplexity of three pruning metrics across different sparsity ratios. We observe that Pruner-Zero consistently outperforms other metrics, which demonstrates that careful designs for weight importance metrics can further improve the resulting sparsity pattern.

Table 5: Training costs for different model sizes.

| Training Cost (Time / GPUs) | | | |
|---|---|---|---|
| 6.7B | 7B | 8B | 13B |
| 37.7 min / 2× | 42.5 min / 2× | 60.5 min / 2× | 2 hours / 4× |

Table 6: Impact of hyperparamter $\lambda_{reg}$

| LLaMA-2-7B WikiText2 Perplexity (PPL) Dense:5.47 | | | | | | |
|---|---|---|---|---|---|---|
| $\lambda_{reg}$ | 2.0 | 4.0 | 8.0 | 12.0 | 16.0 | 20.0 | 24.0 |
| PPL | NC | NC | NC | 13.18 | 12.92 | 13.11 | 12.96 |

## 6 Conclusion

In this paper, we propose Lua-LLM, a gradient-based global pruning framework that learns unstructured sparsity allocation for large language models. Lua-LLM splits the global mask selection problem into multiple row-wise subproblems, and leverages a soft Top-K operator to generate differentiable pruning masks for each row with a single learnable threshold, which enables efficient end-to-end optimization for the model performance. Extensive experiments show that Lua-LLM outperforms existing methods in perplexity and accuracy, especially at high sparsity levels. Our pruning framework achieves adaptive allocation for both layer-wise and intra-layer sparsity, which reveals the inherent sparsity distribution in LLMs and enhances their practical applicability under high sparsity levels.

## 7 Limitation and Future Work

While Lua-LLM achieves significant improvements in model performance over existing methods, there are still some limitations for our methods. First, the end-to-end sparsity learning process for LLMs requires sufficient GPU resources to fulfill the training efficiency, and the learning-based pruning approach is computationally more expensive than metric-based one-shot pruning. Second, similar to other adaptive unstructured pruning methods, the element-wise sparsity pattern requires specific inference frameworks to achieve practical speedup on GPUs, and the SpInfer framework employed in our paper still faces limitations during the prefill phase when batch size and sequence length are large, which leads to higher memory access overhead for the SpMM operations. Future work will explore flexible one-shot pruning methods and advanced techniques for optimizing inference performance for sparse LLMs.

## Acknowledgement

This research was supported by the 2023 Top-Notch Student Training Program 2.0 for Basic Disciplines (20231008).

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

# A Adaptive Unstructured Sparsity Allocation

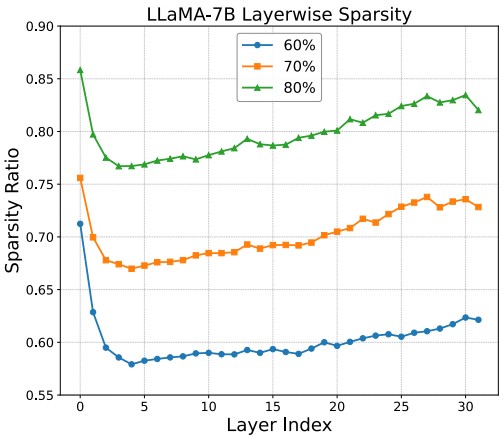
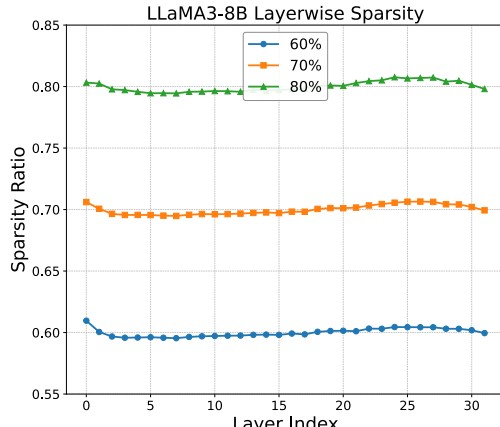

Figure 8: Layer-wise sparsity distribution for LLaMA-7B.

Figure 9: Layer-wise sparsity distribution for LLaMA3-8B.

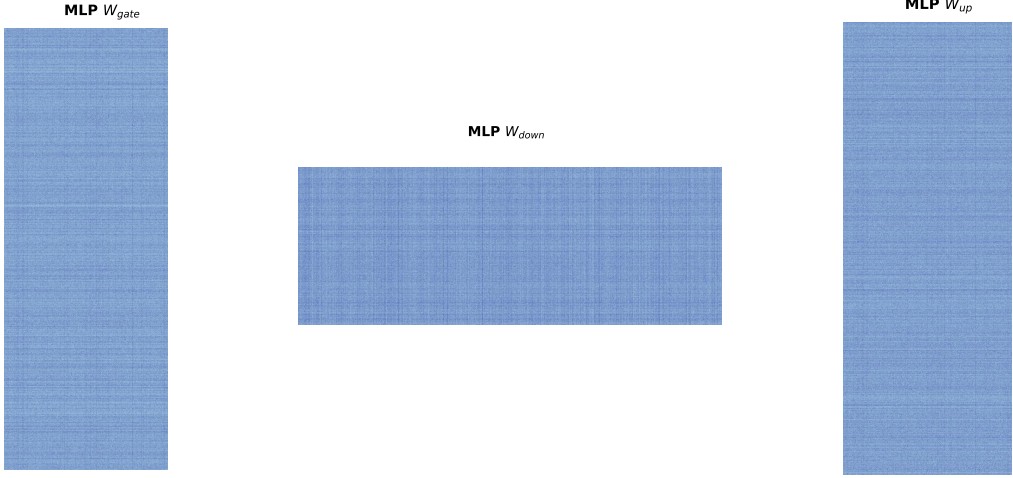

Figure 10: Intra-layer sparsity allocation for MLP layers in the 70% sparse LLaMA2-7B model.

# B Additional Ablation Studies

**Pruning Granularity.**  To verify the scalability of the observation in Wanda that row-wise pruning is the optimal choice for weight importance comparison, we conduct an ablation experiment under different pruning granularities, i.e. different comparison group selection strategies. The results in Table 7 demonstrate that row-wise comparison remains the optimal choice for the adaptive sparsity allocation scenario, aligning with Wanda's observations for the uniform pruning strategy, which is discussed in Section 3.2 of our paper.

**Training Datasets Size and Domain.**  Furthermore, we explore the sensitivity of our method to the training dataset used (e.g., the size of the training set and its domain/distribution). We conduct an ablation study on the LLaMA2-7B model with different size of C4 training dataset across different sparsity levels. The results demonstrate that the performance improves with larger training datasets.

Table 7: Perplexity of sparse LLaMA2-7B on WikiText-2 under different pruning granularities.

| Granularity | 50% | 60% | 70% | 80% |
|---|---|---|---|---|
| Row-wise | **6.85** | **8.54** | **12.92** | **30.27** |
| Column-wise | 7.16 | 9.42 | 14.17 | 51.76 |
| Layer-wise | 7.32 | 14.40 | 210.48 | 28599.64 |

Table 8: Perplexity on Wikitext2 with different training samples and domains.

| Samples | 128 | 256 | 512 | 1024 |
|---|---|---|---|---|
| 70% | 60.82 | 22.52 | 16.05 | 12.92 |
| 80% | 211.17 | 68.38 | 38.46 | 30.27 |

| Datasets | 50% | 60% | 70% |
|---|---|---|---|
| C4 | 6.85 | 8.54 | 12.92 |
| WikiText103 | 6.20 | 7.45 | 10.57 |

We also conduct an ablation study on the LLaMA2-7B model with C4 and WikiText103 datasets across different sparsity levels. The results in Table 8 reveal that our method on two different training datasets achieve similar performance, demonstrating the robustness and generalization capability of our method across different datasets. Moreover, we notice that the perplexity of the sparsity pattern learned from WikiText103 is better than C4. A potential reason is that the WikiText103 training dataset has a more similar distribution to the validation data, which is obtained from WikiText2.

**Speedups for Different Pruning Methods.** We evaluate the inference speedups of Wanda and Lua-LLM for OPT-6.7B model at 50%-70% sparsity levels. The results in Table 9 shows that the speedups of different pruning methods are quite the same for each sparsity level, which is compatible with the parameter counts. The results demonstrate that although adaptive unstructured pruning methods introduce a more irregular pattern for the sparse weight matrices, we can achieve meaningful speedup for the overall model with multiple performance optimization techniques employed in SpMM kernels like SpInfer, such as efficient sparse format and fine-grained execution pipeline.

**Relative Order for Importance Metric.** We conduct an ablation study on LLaMA2-7B (Multi-Head Attention architecture) and Mistral-7B (Grouped-Query Attention architecture) models at 70% sparsity with different importance metrics (Magnitude, Wanda), and allocation strategies (with or without Lua-LLM). The results in Table 10 demonstrate that: (1) When integrated with Lua-LLM, the relative order among Magnitude and Wanda importance metrics is preserved under different models and metrics. (2) For different importance metrics and model architectures, our Lua-LLM consistently outperforms the uniform pruning baseline Wanda.

Table 9: End-to-end inference throughput (token/s) speedups for different pruning methods.

| Method | Sparsity | Dense | 50% | 60% | 70% |
|---|---|---|---|---|---|
| Wanda | Throughput ↑ | 696.6 | 844.9 | 1013.5 | 1200.6 |
| | Speedup ↑ | - | 1.21× | 1.45× | 1.72× |
| Lua-LLM | Throughput ↑ | 696.6 | 842.8 | 1012.7 | 1202.3 |
| | Speedup ↑ | - | 1.21× | 1.45× | 1.73× |

Table 10: Perplexity evaluation for importance metrics, model architectures and allocation strategies.

| Metric | Magnitude | | Wanda | |
|---|---|---|---|---|
| Model | w/o Lua-LLM | w/ Lua-LLM | w/o Lua-LLM | w/ Lua-LLM |
| LLaMA2-7B | 141643 | 24.14 | 75.42 | 12.92 |
| Mistral-7B | 221.88 | 16.10 | 60.03 | 11.54 |

# C Additional Evaluation Results

## C.1 Zero-shot Accuracy Evaluation

Table 11: Zero-shot accuracy ↑ results on seven downstream tasks for the pruned LLaMA-7B, LLaMA2-7B and LLaMA3-8B models at 60% sparsity level.

| Model | Method | BoolQ ↑ | PIQA ↑ | HellaSwag ↑ | WinoGrande ↑ | ARC-e ↑ | ARC-c ↑ | OBQA ↑ | Mean ↑ |
|---|---|---|---|---|---|---|---|---|---|
| LLaMA-7B | Dense | 73.12 | 78.67 | 56.41 | 67.09 | 67.30 | 38.31 | 28.20 | 58.44 |
| | Wanda | 67.92 | 72.20 | 43.45 | 59.75 | 56.57 | 29.95 | 24.20 | 50.58 |
| | OWL | 69.30 | 72.80 | 46.21 | 61.88 | 56.36 | 31.57 | 25.20 | 51.90 |
| | DSA | 68.26 | 72.52 | 45.97 | 61.88 | 54.46 | 31.83 | 24.40 | 51.33 |
| | Lua-LLM | **70.89** | **73.88** | **52.82** | **64.17** | **62.04** | **35.24** | **26.20** | **55.03** |
| LLaMA2-7B | Dense | 71.13 | 78.07 | 56.69 | 67.17 | 69.28 | 39.93 | 31.60 | 59.12 |
| | Wanda | 65.29 | 71.70 | 43.77 | 64.17 | 64.60 | 30.97 | 25.80 | 52.33 |
| | OWL | 66.85 | 72.74 | 46.64 | **66.77** | **67.76** | 32.34 | 27.60 | 54.39 |
| | DSA | **71.71** | 72.63 | 45.39 | 65.75 | 65.32 | 31.91 | 28.80 | 54.50 |
| | Lua-LLM | 70.28 | **73.29** | **52.69** | 65.51 | 65.70 | **37.29** | **30.00** | **56.39** |
| LLaMA3-8B | Dense | 81.25 | 79.71 | 60.18 | 72.69 | 80.09 | 50.51 | 34.80 | 65.60 |
| | Wanda | 68.10 | 67.95 | 37.75 | 60.30 | 59.51 | 27.56 | 20.00 | 48.74 |
| | OWL | 71.22 | 70.95 | 41.65 | 64.01 | 62.29 | 31.66 | 23.40 | 52.17 |
| | DSA | 65.32 | 68.88 | 37.08 | 60.06 | 60.52 | 27.13 | 22.00 | 48.71 |
| | Lua-LLM | **75.41** | **73.12** | **44.32** | **70.48** | **63.11** | **32.06** | **28.60** | **55.30** |

Table 12: Zero-shot accuracy ↑ results on seven downstream tasks for the pruned LLaMA-7B, LLaMA2-7B and LLaMA3-8B models at 80% sparsity level.

| Model | Method | BoolQ ↑ | PIQA ↑ | HellaSwag ↑ | WinoGrande ↑ | ARC-e ↑ | ARC-c ↑ | OBQA ↑ | Mean ↑ |
|---|---|---|---|---|---|---|---|---|---|
| LLaMA-7B | Dense | 73.12 | 78.67 | 56.41 | 67.09 | 67.30 | 38.31 | 28.20 | 58.44 |
| | Wanda | 37.86 | 53.43 | 26.44 | 48.38 | 26.56 | 20.82 | 13.40 | 32.41 |
| | OWL | 49.82 | 53.70 | 26.54 | 50.67 | 26.35 | 19.71 | 11.00 | 33.97 |
| | DSA | 37.83 | 54.08 | 26.68 | 49.80 | 27.61 | 20.73 | 10.40 | 32.45 |
| | Lua-LLM | **61.71** | **64.36** | **35.03** | **51.78** | **40.70** | **22.35** | **17.80** | **41.96** |
| LLaMA2-7B | Dense | 71.13 | 78.07 | 56.69 | 67.17 | 69.28 | 39.93 | 31.60 | 59.12 |
| | Wanda | 37.83 | 52.61 | 25.87 | 48.54 | 26.64 | 19.54 | 13.40 | 32.06 |
| | OWL | 37.86 | 54.35 | 26.43 | 49.96 | 27.65 | 19.28 | 13.00 | 32.65 |
| | DSA | 37.83 | 53.86 | 26.38 | 49.09 | 27.06 | 19.80 | 12.00 | 32.29 |
| | Lua-LLM | **61.38** | **61.32** | **32.18** | **55.01** | **38.93** | **19.62** | **17.80** | **40.89** |
| LLaMA3-8B | Dense | 81.25 | 79.71 | 60.18 | 72.69 | 80.09 | 50.51 | 34.80 | 65.60 |
| | Wanda | 37.83 | 52.77 | 26.52 | 49.64 | 28.19 | 19.37 | 10.80 | 32.16 |
| | OWL | 42.29 | 53.54 | 26.80 | 47.75 | 28.32 | 20.48 | 13.60 | 33.25 |
| | DSA | 37.83 | 53.26 | 26.56 | 48.70 | 27.99 | 19.20 | 16.60 | 32.88 |
| | Lua-LLM | **58.10** | **58.38** | **28.73** | **51.07** | **34.22** | **18.77** | **14.00** | **37.47** |

## C.2 Few-shot Knowledge Reasoning Evaluation

We evaluate the 5-shot accuracy on the MMLU dataset for LLaMA-7/13B and LLaMA2-7/13B models pruned by Wanda, OWL and Lua-LLM, with 50%-70% sparsity levels as well as the dense baselines. The results show that:

(1) Compared to the dense LLaMA2-7B model (45.8%), the 60% sparse LLaMA2-13B model pruned with Lua-LLM demonstrates superior performance (46.6%), while Wanda (35.3%) and OWL (40.4%) cannot achieve this superiority at 60% sparsity level. A similar trend is observed for the LLaMA-V1 pair (35.1% vs. 38.4%). These findings demonstrate that in terms of the so-called "large+sparse vs. small+dense" comparison, Lua-LLM achieves useful improvements at the higher sparsity level, which enhances its practical benefits.

(2) For larger models with 13B parameters, the performance gains of Lua-LLM are much larger than that of smaller models. This indicates that when pruning the smaller models (e.g., with 7B

Table 13: 5-shot MMLU accuracies (%) across different models, sparsity levels, and methods.

| Method | Sparsity | LLaMA-7B | LLaMA-13B | LLaMA2-7B | LLaMA2-13B |
|---|---|---|---|---|---|
| Dense | 0% | 35.1 | 47.0 | 45.8 | 55.7 |
| Wanda | | 30.5 | 38.7 | 34.2 | 48.3 |
| OWL | 50% | 30.9 | 40.6 | 33.6 | 48.7 |
| Lua-LLM | | **31.3** | **41.2** | **34.8** | **49.4** |
| Wanda | | 27.2 | 31.8 | 28.9 | 35.3 |
| OWL | 60% | 28.5 | 34.3 | 30.3 | 40.4 |
| Lua-LLM | | **30.1** | **38.4** | **32.1** | **46.6** |
| Wanda | | 24.6 | 25.3 | 24.5 | 26.8 |
| OWL | 70% | 25.8 | 27.3 | 25.6 | 28.2 |
| Lua-LLM | | **26.7** | **30.6** | **26.5** | **32.0** |

parameters), their MMLU accuracies are relatively harder to preserve. Moreover, OWL faces performance degradation (33.6%) compared to Wanda (34.2%) for the 50% sparse LLaMA2-7B model, while Lua-LLM (34.8%) consistently outperforms the Wanda baseline, which demonstrates the robustness of our method across different sparsity levels and models.

## C.3 Integrate with Post-training Pruning Techniques

Lua-LLM is compatible with post-training pruning techniques like SparseGPT. Although SparseGPT iteratively selects block-wise masks and updates the other unpruned weights to compensate the pruning error, which is expensive to integrate into the training process, we can use our Lua-LLM to search for block-wise sparsity, and then apply the sparsity allocation to SparseGPT.

To verify this, we adopt a block-wise mask selection granularity for Lua-LLM, and search for the optimal sparsity allocation for these intra-layer groups. After the learning process, we consider the searched sparsity allocation as the block-wise sensitivity statistics to weight pruning, and apply it to SparseGPT, which uses the searched sparsity to select block-wise masks and updates the other weights. The results show that our allocation strategy achieves superior performance at different sparsity levels. However, the results also demonstrate that the weight update strategy adopted by SparseGPT is not sufficient to recover model performance at higher sparsity ratios, which underscores the importance of mask selection granularity.

Table 14: Wikitext2 perplexity of sparse LLaMA2-7B models pruned with SparseGPT.

| Method | 50% | 60% | 70% | 80% |
|---|---|---|---|---|
| SparseGPT | 6.99 | 10.18 | 28.50 | 113.36 |
| SparseGPT w. OWL | 6.94 | 9.21 | 20.32 | 90.44 |
| **SparseGPT w. Lua-LLM** | **6.83** | **9.05** | **16.14** | **43.63** |

## C.4 Integrate with Fine-tuning Process

**LoRA fine-tuning.** We conduct an ablation study on sparse LLaMA2-7B pruned with Wanda and Lua-LLM for LoRA fine-tuning with Alpaca training dataset. Each model is fine-tuned on 1 A100 GPU for 3 epochs, which takes about 13 hours. We report the training loss and evaluate the model performance after fine-tuning. The results show that: (1) The loss converges in around 3 epochs, ensuring a meaningful comparison. (2) Compared to the final converged training loss of Wanda (in around 13 hours), our Lua-LLM achieves comparable values in 0.5 epoch (in around 2 hours). (3) Compared to Wanda, our Lua-LLM improves model quality after fine-tuning.

**Full Parameter Fine-tuning.** We conduct full parameter fine-tuning on the auxiliary_train subset of MMLU datasets for the sparse weights in 70% sparse LLaMA2-7B models, pruned by Wanda and Lua-LLM, respectively. Each model is fine-tuned on 4 A100 GPUs for 20000 steps (in around 25 hours). The results show that Lua-LLM achieves superior performance compared to Wanda under further fine-tuning, which is consistent with the findings in prior literature that optimizing the mask selection strategy helps to improve the performance of pruned models under re-training/fine-tuning

process. Moreover, the experiment also demonstrates the substantial computational overhead of the full parameter fine-tuning process on large language models, and we would like to explore other cost-effective weight reconstruction strategies with global optimality in our future study.

Table 15: Training loss convergence of sparse LLaMA2-7B during the LoRA fine-tuning process.

| Epoch | 0 | 0.5 | 1 | 1.5 | 2 | 2.5 | 3 |
|---|---|---|---|---|---|---|---|
| Wanda (70%) | 4.09 | 1.21 | 1.16 | 1.14 | 1.13 | 1.12 | 1.12 |
| Lua-LLM (70%) | 2.50 | 1.10 | 1.07 | 1.05 | 1.04 | 1.03 | 1.02 |
| Wanda (80%) | 7.03 | 1.92 | 1.78 | 1.72 | 1.68 | 1.66 | 1.66 |
| Lua-LLM (80%) | 3.10 | 1.42 | 1.35 | 1.32 | 1.31 | 1.29 | 1.28 |

Table 16: MMLU (%) evaluation with and without fine-tuning process.

| Method | MMLU (%) |
|---|---|
| Wanda | 24.5 |
| Wanda w. LoRA FT | 26.8 |
| Wanda w. Full FT | 31.3 |
| Lua-LLM | 26.5 |
| Lua-LLM w. LoRA FT | 30.3 |
| Lua-LLM w. Full FT | 35.6 |

