# OpenReview forum: "Lua-LLM: Learning Unstructured-Sparsity Allocation for Large Language Models"
_NeurIPS.cc/2025/Conference — NeurIPS 2025 poster_

### Official Review · Reviewer_tzDL · 2025-07-01

**Clarity:** 3
**Significance:** 3
**Originality:** 3
**Rating:** 5
**Confidence:** 3

**Summary:**

This paper proposes a new model pruning technique for LLMs. Specifically, instead of uniformly allocating the sparsity ration per layer, or using heuristic statistics like in prior work, the proposed method uses an end-to-end approach to learn the sparsity allocation. Empirical experiments demonstrate promising results at high sparsity with practical speed up observed.

**Questions:**

- Is perplexity a good evaluation metric for comparisons between different pruning strategies? Table 1 suggests that Llama3-8B has higher perplexity than Llama-7B (for both dense models). However, their performances on downstream tasks show the opposite in Table 2.
- Could the authors discuss practical trade-offs between structured and unstructured pruning strategies in terms of end-to-end speed up and downstream performances?
- How sensitive is the proposed method to the training dataset used (e.g., the size of the training set and its domain/distribution)?

**Ethical Concerns:**

["NO or VERY MINOR ethics concerns only"]

**Final Justification:**

The method is well-motivated and the paper is clearly written with promising empirical results.

**Limitations:**

Yes.

**Paper Formatting Concerns:**

No concerns on paper formatting.

**Quality:**

3

**Strengths And Weaknesses:**

Strengths:
- The paper is clearly written. The related work section also gives reader a clear context to LLM pruning literature.
- The proposed method is intuitive and well-motivated. Previous pruning approach adopts uniform sparsity allocation or using heuristic statistics (like outlier ratio as in OWL) to determine sparsity allocation. Instead, it is natural to consider a learning based approach to determine the sparsity ratio as considered in this work. This learning is made possible by employing sigmoid approximation for top-k selection which is intuitive and shown to be effective in the experiments.
- Empirical results are promising, especially at high sparsity ratio. Further ablation studies are done to help better understanding of the proposed method and its comparison to existing methods.

Weaknesses:
- While there are significant improvements on the perplexity evaluations, the improvements on downstream tasks are relatively small. Also, the downstream tasks considered here can be made more comprehensive by adding common evaluation datasets like MMLU.
- The practical speed up observed is not very large even at 70% sparsity ratio. Could the authors discuss whether this limits the practical usage of unstructured pruning methods?

---

> ### Author Rebuttal · Authors · 2025-07-31
>
> We are glad that the reviewer found that our work is clearly written and well-motivated. We thank the reviewer for the constructive feedback and appreciate the opportunity to address the questions you have raised.
>
> ---
>
> > Q1: The downstream tasks can be made more comprehensive by adding common evaluation datasets like MMLU.
>
> > Q3: Is perplexity a good evaluation metric for comparisons between different pruning strategies?
>
> **A1**: We agree with the reviewer that when comparing different types of models, there can be mismatchs between perplexity and downstream performance. However, we think that perplexity remains a straightforward and efficient evaluation metric for comparing different pruning methods **on the same model type**.
>
> To demonstrate more comprehensive comparisons, we evaluate the 5-shot accuracy on the MMLU dataset for LLaMA2-7B and LLaMA3-8B model pruned with Wanda, OWL, and Lua-LLM, at 60% and 70% sparsity levels. The results (**See Table below**) show the superior performance of our method.
>
> *Table 1: Comparisons for 5-shot accuracy on the MMLU dataset.*
>
> | Sparsity | Method  | LLaMA2-7B | LLaMA3-8B |
> | -------- | ------- | --------- | --------- |
> | 60%      | Wanda   | 28.9      | 29.9      |
> |          | OWL     | 30.3      | 34.0      |
> |          | Lua-LLM | **32.1**  | **35.1**  |
> | 70%      | Wanda   | 24.5      | 26.1      |
> |          | OWL     | 25.6      | 27.3      |
> |          | Lua-LLM | **26.5**  | **28.6**  |
>
> > Q2: The practical speedup observed is not very large even at 70% sparsity ratio. Could the authors discuss whether this limits the practical usage of unstructured pruning methods?
>
> **A2**: We thank the reviewer for raising this meaningful discussion for the practical usage of unstructured pruning methods. Our inference evaluation is conducted with SpInfer framework. As discussed in its paper [1], SpInfer faces limitations during the **prefill phase** when batch size and sequence length are large, which leads to higher memory access overhead for the SpMM operations. Considering the relatively low performance degradation and substantial memory savings offered by unstructured pruning methods, optimizing SpMM operations on GPUs remains an important research topic for future work. Furthermore, future research directions include designing hardware-friendly sparsity patterns like V:N:M [2], which can leverage **Sparse** **Tensor** **Cores** on NVIDIA GPUs without significant performance degradation.
>
> > Q4: Could the authors discuss practical trade-offs between structured and unstructured pruning strategies in terms of end-to-end speed up and downstream performances?
>
> **A4**: **Structured** **pruning** removes entire components, such as attention-heads in the Attention layers and neurons in the MLP layers, which can directly generate a smaller "dense" model and achieve significant inference speedup on the GPU Tensor Core architecture. However, it typically suffers severe performance degradation. Even if employing weight reconstruction methods such as LoRA-finetuning [3] and ADMM-based reformation [4], the performance loss is still difficult to compensate, which limits its sparsity levels under 50% typically.
>
> **Unstructured** **pruning** removes less critical weight parameters at the element granularity, which offers the most flexibility and yields better post-training performance than structured pruning even at high sparsity levels. Despite the significant performance gains and memory savings, a major challenge for unstructured pruning is to achieve practical speedups on GPUs with substantial irregular memory access overhead, especially for large language models. Sparse inference framework like SpInfer [1] manages to optimize the SpMM operations for GPU Tensor Core architectures, yet its inference throughput is limited during the prefill stage.
>
> N:M sparsity serves as a trade-off between flexibility and efficiency, yet only 2:4 sparsity is supported by Sparse Tensor Cores on NVIDIA GPUs. Moreover, recent studies have focused on designing more **adaptive sparsity patterns**. For example, DISP-LLM [5] proposes a dimension-independent structured sparsity pattern, which prunes the channels for each weight matrix independently. To enhance the practical usage of model pruning, we will focus on other adaptive sparsity patterns in future exploration.
>
> > Q5: How sensitive is the proposed method to the training dataset used (e.g., the size of the training set and its domain/distribution)?
>
> **A5**: We conduct an **ablation study on the LLaMA2-7B model with different size of C4 training dataset across different sparsity levels**. The results (**See Table below**) demonstrate that the performance improves with larger training datasets. Notably, even with only 128 training samples, Lua-LLM demonstrates the ability to achieve superior results (60.82) compared to Wanda (75.42).
>
> *Table 2:* *Perplexity* *of* *sparse* *LLaMA2-7B models on Wikitext2 with different training samples.*
>
> | samples | 128    | 256   | 512   | 1024  |
> | ------- | ------ | ----- | ----- | ----- |
> | 70%     | 60.82  | 22.52 | 16.05 | 12.92 |
> | 80%     | 211.17 | 68.38 | 38.46 | 30.27 |
>
> We also conduct an **ablation study on the LLaMA2-7B model with C4 and Wikitext103 datasets across different sparsity levels**. The results (**See Table below**) reveal that our method on two different training datasets achieve similar performance, demonstrating the robustness and generalization capability of our method across different datasets.
>
> *Table 3: Performance of* *sparse* *LLaMA2-7B models with different training datasets.*
>
> |             | 50%  |      | 60%  |      | 70%   |      |
> | ----------- | ---- | ---- | ---- | ---- | ----- | ---- |
> |             | PPL  | MMLU | PPL  | MMLU | PPL   | MMLU |
> | C4          | 6.85 | 34.5 | 8.54 | 32.1 | 12.92 | 26.5 |
> | WikiText103 | 6.20 | 34.2 | 7.45 | 31.7 | 10.57 | 26.8 |
>
> ---
>
> Thanks for the attentive reading of the manuscript and constructive feedback. We hope our response addresses all the concerns and that the reviewer will consider raising the rating accordingly. We are glad to discuss further questions and suggestions.
>
> ---
>
> [1] Fan, Ruibo, et al. "Spinfer: Leveraging low-level sparsity for efficient large language model inference on gpus." *Proceedings of the Twentieth European Conference on Computer Systems*. 2025.
>
> [2] Castro, Roberto L., et al. "Venom: A vectorized n: M format for unleashing the power of sparse tensor cores." *Proceedings of the International Conference for* *High Performance Computing**, Networking, Storage and Analysis*. 2023.
>
> [3] Ma, Xinyin, Gongfan Fang, and Xinchao Wang. "Llm-pruner: On the structural pruning of large language models." *Advances in neural information processing systems* 36 (2023): 21702-21720.
>
> [4] Shen, Xuan, et al. "Search for efficient large language models." *Advances in Neural Information Processing Systems* 37 (2024): 139294-139315.
>
> [5] Gao, Shangqian, et al. "Disp-llm: Dimension-independent structural pruning for large language models." *Advances in Neural Information Processing Systems* 37 (2024): 72219-72244.

---

> > ### Comment · Reviewer_tzDL · 2025-08-04
> >
> > Thanks to the authors for the response. They have addressed my questions, and I would like to keep my original rating.

---

### Official Review · Reviewer_wteP · 2025-07-02

**Clarity:** 3
**Significance:** 2
**Originality:** 3
**Rating:** 5
**Confidence:** 5

**Summary:**

Unstructured sparsity's generality is great for maintaining model quality when pruning, but this generality also complicates deciding where to put the sparsity.  This submission tackles this problem by directly learning the optimal sparsity for each row of the model's weights.  Each row is initialized with uniform sparsity (given some target p), and the mask is chosen with Wanda's importance score.  The mask is then optimized via gradient descent; through this process, the sparsity in each row is adjusted via a soft top-k operator given the learnable threshold and importance scores for each weight.  A regularization term seeks to control the model's overall sparsity, while standard language modeling loss adjusts the sparsity per-row.  Empirical evaluation shows modest perplexity benefit compared to baselines at 50% sparsity, but improvements are pronounced at 80% sparsity, and zero-shot accuracy shows meaningful improvements at 70% sparsity.  The authors performed ablation studies to demonstrate the importance of searching for the right sparsity distribution, their regularization term, and how different importance metrics affect final model quality.

**Questions:**

1. What batch size and sequence lengths were used for inference speedups in table 4?
2. Is Lua-LLM robust to different importance metrics?  The ablation study shows that different importance metrics behave differently for one model and metric, but this does not show that the relative ordering among importance metrics is preserved under different models or metrics, or that Lua-LLM improves the results of each importance metrics without Lua-LLM.
- Regularization:
3. What was the final sparsity achieved by each model in Table 6?  (Does regularizer strength affect sparsity, or just mask quality?)
4. What does it look like if this regularization term "doesn't converge" (line 322)?
- Behavior under fine-tuning: the results presented aren't strong enough to convince me to use unstructured sparsity without adjusting weights to compensate for those that are removed.  Two questions follow:
5. Is Lua-LLM inherently incompatible with techniques like SparseGPT?  It seems like it may be too computationally expensive to re-compute weight importances with changing masks, since other weights are updated as part of the sparsification process.
6. Are these layer-wise sparsity reallocations necessary in the face of continued pre-training or fine-tuning?  If post-training techniques aren't sufficient to recover quality, and weight adjustments are necessary, it is important to show that improving the mask selection meaningfully improves model quality after weights are adjusted.

The most important questions are (6) and (5).  Questions (1), (3), and (4) are important for the submission's quality, and (2) is nice-to-have, but not crucial.

**Ethical Concerns:**

["NO or VERY MINOR ethics concerns only"]

**Final Justification:**

The authors addressed all of my concerns, and I'm now confident that the method is well-founded theoretically and can provide meaningful results in a wide range of scenarios as shown by the new empirical results provided during the rebuttal phase.

**Limitations:**

Unstructured sparsity is notoriously difficult to accelerate on GPUs.  The submission relies on SpInfer, which itself noted that when the batch size * sequence length is "large" (as is often the case in the prefill phase of model deployment), it becomes slower than dense operation due to the overheads.
If my understanding is correct, Lua-LLM is incompatible with SparseGPT and other techniques that adjust the weight values themselves.

**Paper Formatting Concerns:**

No concerns.

**Quality:**

2

**Strengths And Weaknesses:**

## Strengths
This submission's strengths are its clarity and originality.
- Clarity: I generally found the paper easy to read with clear motivation.  It answered most of my questions in the section after they occurred to me, and I feel like I could likely reproduce the results given the descriptions and equations, each of which was helpful.
- Originality: I believe the ideas within are novel in this context -- soft top-k is not new, but its use here is unique, and the regularization to control overall sparsity seems like an elegant solution.

## Weaknesses
Its main weakness is in its significance.  While it does improve upon the baselines, Lua-LLM does not show that it's preferable to use one of these sparse models over a smaller, dense counterpart.  Take, for example, LLaMa2-7B and LLaMa2-13B in Table 1.  The dense 7B model has a perplexity of 5.47, and the 50% sparse 13B model, ostensibly around the same size, has a perplexity slightly higher, 5.89.  A similar trend is seen in LlAMA (1) 7B and 13B and OPT 6.7B and 13B.  Why should a user bother to start with a large version of a model and apply unstructured sparsity to achieve a smaller model, when directly training a smaller model gives similar (or better) quality?  Further, the speedups reported for the OPT models suggest that OPT-13B at 70% sparsity achieves only slightly lower throughput than the dense 6.7B model, but its perplexity is significantly worse - 20.96 vs. 10.86.


I also have relatively minor qualms about the submission's quality.  Quality is otherwise good - claims are generally supported by evaluations, and while the limitations section should really be in the main text, I'll trust that this can be accommodated if the submission is ultimately accepted.  My qualms:
- Evaluation is limited to only two model families: LLaMA and OPT.  Though different LLaMA versions were used, the results would be more compelling with a broader set of models with more varying architectures.
- The regularization term's benefit is not clear.  What does it look like if this term "doesn't converge" (line 322)?  Does the final overall sparsity of the model change with regularization strength?
- Details for the inference speedups are missing - what batch size and sequence lengths (input/output) are used?

---

> ### Author Rebuttal · Authors · 2025-07-31
>
> We are glad that the reviewer found that our work is clearly motivated and novel. We thank the reviewer for the constructive feedback and appreciate the opportunity to address the questions you have raised.
>
> ---
>
> > Q1: Significance concerns: Lua-LLM does not show that it's preferable to use one of these sparse models over a smaller, dense counterpart.
>
> **A1**: We thank the reviewer for raising this meaningful discussion regarding performance comparisons between large sparse LLMs and small dense LLMs with similar parameter counts.
>
> First, we would like to clarify that large sparse LLMs with unstructured sparsity are often better than small dense LLMs with similar parameter counts on **downstream task performance**, even without any weight updates, which is also discussed in Section 4.1 of Wanda's paper [1]. To verify the practical usage of unstructured pruning methods without weight updates, we **evaluate the average zero-shot accuracies on 7 downstream tasks for the LLaMA2-13B model at 50% sparsity level with Wanda, OWL, and Lua-LLM**. The results (**See Table below**) demonstrate that unstructured 50% sparse LLaMA2-13B with OWL (60.03%) outperforms dense LLaMA2-7B (59.12%), and our Lua-LLM further improve the accuracy to 60.91%.
>
> *Table 1: Performance comparisons between large sparse LLMs and small dense LLMs*
>
> | Model                    | Zero-shot Accuracy |
> | ------------------------ | ------------------ |
> | LLaMA2-7B (Dense)        | 59.12              |
> | LLaMA2-13B (Dense)       | 61.84              |
> | LLaMA2-13B (Wanda 50%)   | 58.47              |
> | LLaMA2-13B (OWL 50%)     | 60.03              |
> | LLaMA2-13B (Lua-LLM 50%) | 60.91              |
>
> Moreover, we understand the reviewer's concern that the 50% sparse LLaMA2-13B model has a higher perplexity than the dense LLaMA2-7B model, yet a potential reason for this mismatch is that the perplexity metric cannot reflect the **practical performance difference between different structures of models**. For example, LLaMA3-8B shows superior zero-shot accuracy than LLaMA2-7B (65.60% vs. 59.12%), but it has a larger perplexity than LLaMA2-7B (6.14 vs. 5.47). Notably, perplexity remains a straightforward and efficient evaluation metric for **comparing different pruning methods on the same model type**.
>
> *Table 2: Performance comparisons with perplexity and downstream accuracy.*
>
> | Model     | PPL      | Zero-shot Accuracy |
> | --------- | -------- | ------------------ |
> | LLaMA2-7B | **5.47** | 59.12              |
> | LLaMA3-8B | 6.14     | **65.60**          |
>
> > Q2: Evaluation is limited to only two model families: LLaMA and OPT.
>
> > Q6: Is Lua-LLM robust to different importance metrics?
>
> **A2**: We conduct an **ablation study on LLaMA2-7B (Multi-Head Attention architecture) and Mistral-7B (Grouped-Query Attention architecture) models at 70% sparsity with different importance metrics (Magnitude, Wanda), evaluation metrics (PPL, MMLU), and allocation strategies (with or without Lua-LLM)**. The results (**See Table below**) demonstrate that:
>
> (1) When integrated with Lua-LLM, the relative order among Magnitude and Wanda importance metrics is preserved under different models and metrics.
>
> (2) For different importance metrics, model architectures, and evaluation metrics, our Lua-LLM consistently outperforms the uniform pruning baseline Wanda.
>
> *Table 3: Robustness evaluation for importance metrics, evaluation metrics, and allocation strategies.*
>
> |                       | LLaMA2-7B |          | Mistral-7B |          |
> | --------------------- | --------- | -------- | ---------- | -------- |
> |                       | PPL       | MMLU (%) | PPL        | MMLU (%) |
> | Magnitude w/o Lua-LLM | 141643    | 23.8     | 221.88     | 24.2     |
> | Magnitude w/ Lua-LLM  | 24.14     | 25.4     | 16.10      | 25.8     |
> | Wanda w/o Lua-LLM     | 75.42     | 24.5     | 60.03      | 24.9     |
> | Wanda w/ Lua-LLM      | 12.92     | 26.6     | 11.54      | 27.2     |
>
> > Q3: The regularization term's benefit is not clear.
>
> > Q7: What was the final sparsity achieved by each model in Table 6?
>
> > Q8: What does it look like if this regularization term "doesn't converge" (line 322)?
>
> **A3**: We thank the reviewer for raising this meaningful discussion about the effectiveness of the regularization term. If the regularization hyperparameter is not large enough, **the regularization loss fails to converge to zero**. This indicates that the pruning mask learned by our method does not meet the target sparsity, and **the overall sparsity value is actually lower than the target one**, since this can lead to better model performance and lower training loss.  We consider the case as "doesn't converge", since it leads to an **unfair comparison**. When the regularization hyperparameter is sufficiently large, the regularization loss rapidly converges to zero, which strictly enforces the target overall sparsity.
>
> > Q4&Q5: What batch size and sequence lengths (input/output) are used?
>
> **A4**: We use the testcase with batch size set to 8, input sequence length set to 32, and output sequence length set to 256. In this case, the inference throughput of SpInfer is not bottlenecked by the prefill stage, and the sparse model integrated can achieve practical speedup compared to the dense baseline. SpInfer faces limitations during the prefill phase when batch size and sequence length are large, which leads to higher memory access overhead for the SpMM operations. A potential solution [2] [3] is to extract dense blocks of non-zeros in the sparse weight matrices and use dense matrix-matrix multiplication to achieve high throughput, and we plan to leave this for our future study.
>
> > Q9: Is Lua-LLM inherently incompatible with techniques like SparseGPT?
>
> **A9**: Lua-LLM is compatible with techniques like SparseGPT. Although SparseGPT iteratively selects block-wise masks and updates the other unpruned weights to compensate the pruning error, which is expensive to integrate into the training process, we can use our Lua-LLM to **search for block-wise sparsity, and then apply the sparsity allocation to the pruning process of SparseGPT**.
>
> To verify this, we adopt a block-wise mask selection granularity for Lua-LLM, and search for the optimal sparsity allocation for these intra-layer groups. After the learning process, we consider the searched sparsity allocation as the block-wise sensitivity statistics to weight pruning, and apply it to SparseGPT, which uses the searched sparsity to select block-wise masks and updates the other weights. The results (**See Table below**) show that our allocation strategy achieves superior performance at different sparsity levels. However, the results also demonstrate that the weight update strategy adopted by SparseGPT is not sufficient to recover model performance at higher sparsity ratios, which underscores the importance of mask selection granularity. In **A10**, we further show that improving the mask selection meaningfully improves model quality after fine-tuning.
>
> *Table 4: Wikitext2* *perplexity* *of* *sparse* *LLaMA2-7B models pruned with SparseGPT.*
>
> |                      | 50%      | 60%      | 70%       | 80%       |
> | -------------------- | -------- | -------- | --------- | --------- |
> | SparseGPT            | 6.99     | 10.18    | 28.50     | 113.36    |
> | SparseGPT w. OWL     | 6.94     | 9.21     | 20.32     | 90.44     |
> | SparseGPT w. Lua-LLM | **6.83** | **9.05** | **16.14** | **43.63** |
>
> > Q10: Are these layer-wise sparsity reallocations necessary in the face of continued pre-training or fine-tuning?
>
> **A10**: We understand the reviewer's concern with the necessity of improving the mask selection process when considering further fine-tuning. Thus, we conduct an **ablation study on** **sparse** **LLaMA2-7B pruned with Wanda and Lua-LLM for** **LoRA** **fine-tuning with Alpaca training dataset**. Each model is fine-tuned on 1 A100 GPU for 3 epochs, which takes about 13 hours. We report the training loss and evaluate the model performance after fine-tuning. The results (**See Table below**) show that:
>
> (1) The loss converges in around 3 epochs, ensuring a meaningful comparison.
>
> (2) Compared to the final converged training loss of Wanda (in around 13 hours), our Lua-LLM achieves comparable values in 0.5 epoch (in around 2 hours).
>
> (3) Compared to Wanda, our Lua-LLM improves model quality after fine-tuning.
>
> *Table 5: Training loss* *convergence* *of* *sparse* *LLaMA2-7B during the* *LoRA* *fine-tuning process.*
>
> | Epoch         | 0    | 0.5  | 1    | 1.5  | 2    | 2.5  | 3    |
> | ------------- | ---- | ---- | ---- | ---- | ---- | ---- | ---- |
> | Wanda (70%)   | 4.09 | 1.21 | 1.16 | 1.14 | 1.13 | 1.12 | 1.12 |
> | Lua-LLM (70%) | 2.5  | 1.10 | 1.07 | 1.05 | 1.04 | 1.03 | 1.02 |
> | Wanda (80%)   | 7.03 | 1.92 | 1.78 | 1.72 | 1.68 | 1.66 | 1.66 |
> | Lua-LLM (80%) | 3.10 | 1.42 | 1.35 | 1.32 | 1.31 | 1.29 | 1.28 |
>
> *Table 6: Wikitext2* *perplexity* *of fine-tuned LLaMA2-7B models for different* *pruning* *methods.*
>
> | Method  | 70%   | 80%   |
> | ------- | ----- | ----- |
> | Wanda   | 13.50 | 56.27 |
> | Lua-LLM | **11.23** | **26.48** |
>
> ---
>
> Thanks for the attentive reading of the manuscript and constructive feedback. We hope our response addresses all the concerns and that the reviewer will consider raising the rating accordingly. We are glad to discuss further questions and suggestions.
>
> ---
>
> [1] Sun, Mingjie, et al. "A simple and effective pruning approach for large language models." *arXiv* *preprint* *arXiv:2306.11695* (2023).
>
> [2] Rumi, Masuma Akter, et al. "Accelerating sparse cnn inference on gpus with performance-aware weight pruning." *Proceedings of the* *ACM* *International Conference on Parallel Architectures and Compilation Techniques*. 2020.
>
> [3] Yang, Tao, et al. "Spmmplu: A compiler plug-in with sparse ir for efficient sparse matrix multiplication." *2023 60th* *ACM*/*IEEE* *Design Automation Conference (DAC)*. IEEE, 2023.

---

> > ### Comment · Reviewer_wteP · 2025-08-04
> >
> > I'd like to thank the authors for their hard work in preparing robust and useful responses to all reviewers, and I believe the inclusion of this information in the submission will serve to strengthen it.
> >
> > I find that the story around evaluations is still confusing.  I can accept that PPL is best used to compare different pruning techniques within the same model, etc., and I can accept that trends in downstream tasks may be different.  However:
> > 1. Useful improvements large+sparse vs. small+dense behavior is not demonstrated.
> > 2. MMLU results (provided both here and for Reviewer tzDL) are missing the dense baseline results and, after 70% sparsity, regress nearly to what amounts to random guessing on a multiple (four) choice quiz, so I'm unable to really grasp the importance of Lua-LLM.
> >
> > Regarding (1), I appreciate your pointing out that 50% sparse Llama2-13B is competitive with a dense Llama2-7B in some zero-shot tasks.  However, the improvement provided by Lua-LLM over OWL alone is very small in this sparsity regime, and all reviewers noted that Lua-LLM's improvements were particularly notable at high sparsities.  At these higher sparsities where Lua-LLM is more beneficial, does the same large+sparse vs. small+dense story appear?  To support such a claim, I'd hope to see a table like Table 1 in the original submission, but with downstream tasks rather than PPL.  Table 2 shows large enough regressions that, while I can appreciate Lua-LLM improves over the baselines compared, it's not apparent that it's *useful.*
> >
> > Again, though, I appreciate the rest of the authors' responses thus far!

---

> > > ### Author Response · Authors · 2025-08-05
> > >
> > > Dear Reviewer wteP:
> > >
> > > We sincerely appreciate your thoughtful and constructive feedback. In response to your questions and suggestions, we add the following evaluation:
> > >
> > > We **evaluate the 5-shot accuracy on the MMLU dataset for LLaMA-7/13B and LLaMA2-7/13B models pruned by Wanda, OWL and Lua-LLM, with 50%-70% sparsity levels as well as the dense baselines**. The results (See Table 7, 8 below) show that:
> > >
> > > (1) **Compared to the dense LLaMA2-7B model (45.8%),** **the 60%** **sparse** **LLaMA2-13B model pruned with Lua-LLM demonstrates superior performance (46.6%), while Wanda (35.3%) and OWL (40.4%) cannot achieve this superiority at 60% sparsity level**. A similar trend is observed for the LLaMA-V1 pair (35.1% vs. 38.4%). These findings demonstrate that in terms of the so-called “large+sparse vs. small+dense” comparison, **Lua-LLM achieves *useful* improvements at the higher sparsity level, which enhances its practical benefits**.
> > >
> > > (2) For larger models with 13B parameters, the performance gains of Lua-LLM are much larger than that of smaller models. This indicates that **when** **pruning** **the smaller models (e.g., with 7B parameters), their MMLU accuracies are relatively harder to preserve**. Moreover, OWL faces performance degradation (33.6%) compared to Wanda (34.2%) for the 50% sparse LLaMA2-7B model, while Lua-LLM (34.8%) consistently outperforms the Wanda baseline, which **demonstrates the robustness of our method across different sparsity levels and models**.
> > >
> > > (3) Compared to Wanda, our Lua-LLM **achieves a larger improvement on the MMLU accuracies of LLaMA2-7B after** **LoRA** **fine-tuning (on alpaca training datasets) at 70% sparsity levels** (See Table 8 below), which demonstrates the practical benefits of the adaptive mask selection strategy in our method at the extreme sparsity level.
> > >
> > > *Table 7: Evaluation for 5-shot MMLU accuracies (%) across different models, sparsity levels, and methods*
> > >
> > > | Method  | Sparsity | LLaMA-7B | LLaMA-13B | LLaMA2-7B | LLaMA2-13B |
> > > | ------- | -------- | -------- | --------- | --------- | ---------- |
> > > | Dense   | 0%       | 35.1     | 47.0      | 45.8      | 55.7       |
> > > | Wanda   | 50%      | 30.5     | 38.7      | 34.2      | 48.3       |
> > > | OWL     |          | 30.9     | 40.6      | 33.6      | 48.7       |
> > > | Lua-LLM |          | **31.3** | **41.2**  | **34.8**  | **49.4**   |
> > > | Wanda   | 60%      | 27.2     | 31.8      | 28.9      | 35.3       |
> > > | OWL     |          | 28.5     | 34.3      | 30.3      | 40.4       |
> > > | Lua-LLM |          | **30.1** | **38.4**  | **32.1**  | **46.6**   |
> > > | Wanda   | 70%      | 24.6     | 25.3      | 24.5      | 26.8       |
> > > | OWL     |          | 25.8     | 27.3      | 25.6      | 28.2       |
> > > | Lua-LLM |          | **26.7** | **30.6**  | **26.5**  | **32.0**   |
> > >
> > > *Table 8: Evaluation for 5-shot MMLU accuracies (%) of the fine-tuned LLaMA2-7B models (on alpaca training datasets) at 70% sparsity levels.*
> > >
> > > | Method             | MMLU (%) |
> > > | ------------------ | -------- |
> > > | Wanda              | 24.5     |
> > > | Wanda w. LoRA FT   | 26.8     |
> > > | Lua-LLM            | 26.5     |
> > > | Lua-LLM w. LoRA FT | 30.3     |
> > >
> > > We thank the reviewer again for the thoughtful and constructive comments. We hope our response could address the concerns, and that the reviewer will consider raising the rating accordingly. We are glad to discuss further questions and suggestions.
> > >
> > > Best Regards,
> > >
> > > Authors of submission 2963

---

> > > > ### Comment · Reviewer_wteP · 2025-08-06
> > > >
> > > > I appreciate the continued follow-up.  The extra evaluations show that Lua-LLM's improvements are more meaningful for larger models (13B) and higher sparsity (60%).
> > > >
> > > > My other remaining concern, from my original review, is regarding continued pre-training or (added for clarity) _task-specific_ fine-tuning, *not* PEFT (LoRA).  In the latter setting, the sparse weights themselves are not adjusted.
> > > >
> > > > The results provided suggest that LoRA in isolation is insufficient to recover useful quality on Llama2-7B (dense 45.8% -> Lua-LLM 26.5% -> +LoRA FT 30.3%).  In practice, full parameter fine-tuning or continued pre-training is likely to be used to recover quality after the sparsification step.  _Under this setting,_ is Lua-LLM's benefit consistent with the purely post-training setting's benefit?

---

> > > > > ### Author Response · Authors · 2025-08-08
> > > > >
> > > > > Dear Reviewer wteP:
> > > > >
> > > > > We would like to express our sincere gratitude for your time and feedback on our paper. In response to your question, we conduct *full parameter fine-tuning* on the auxiliary_train subset of MMLU datasets for the sparse weights in 70% sparse LLaMA2-7B models, pruned by Wanda and Lua-LLM, respectively. Each model is fine-tuned on 4 A100 GPUs for 20000 steps (in around 25 hours). The results (See Table below) show that Lua-LLM achieves superior performance compared to Wanda under further fine-tuning, which is consistent with the findings in prior literatures [1, 2, 3] that optimizing the mask selection strategy helps to improve the performance of pruned models under re-training/fine-tuning process. Moreover, the experiment also demonstrates the substantial computational overhead of the full parameter fine-tuning process on large language models, and we would like to explore other cost-effective weight reconstruction strategies *with global optimality* in our future study.
> > > > >
> > > > > *Table: Evaluation for 5-shot MMLU accuracies (%) of the fine-tuned LLaMA2-7B models at 70% sparsity levels.*
> > > > >
> > > > > | Method              | MMLU (%) |
> > > > > | ------------------- | -------- |
> > > > > | Wanda               | 24.5     |
> > > > > | Wanda w.  Full FT   | 31.3     |
> > > > > | Lua-LLM             | 26.5     |
> > > > > | Lua-LLM w.  Full FT | 35.6     |
> > > > >
> > > > > We thank the reviewer again for the thoughtful comments. We hope our response could address the concerns, and we promise to include these results in the revised manuscript.
> > > > >
> > > > > Best Regards,
> > > > >
> > > > > Authors of submission 2963
> > > > >
> > > > > ---
> > > > >
> > > > > [1] Frankle, Jonathan, and Michael Carbin. "The lottery ticket hypothesis: Finding sparse, trainable neural networks." *arXiv* *preprint* *arXiv:1803.03635* (2018).
> > > > >
> > > > > [2] Ma, Xinyin, Gongfan Fang, and Xinchao Wang. "Llm-pruner: On the structural pruning of large language models." *Advances in neural information processing systems* 36 (2023): 21702-21720.
> > > > >
> > > > > [3] Sun, Mingjie, et al. "A simple and effective pruning approach for large language models." *arXiv* *preprint* *arXiv:2306.11695* (2023).

---

> > > > > > ### Comment · Reviewer_wteP · 2025-08-08
> > > > > >
> > > > > > This is a fantastic result, thank you for the effort it took to get it; I think it makes your method much more compelling since it meaningfully improves the quality under full parameter fine-tuning of an already well-trained model.
> > > > > >
> > > > > > I'm glad to raise my overall recommendation to "accept" in light of these results (and the authors' prior responses).
> > > > > >
> > > > > > > optimizing the mask selection strategy helps to improve the performance of pruned models under re-training/fine-tuning process
> > > > > >
> > > > > > PS: I could not find evidence in the references you suggested to show this result for *continued pre-training* or fine-tuning.  I'll readily agree that proper mask selection makes a difference when training from scratch, as in [1].  However, [2] uses LoRA (as described in Section 3.3), and [3] does not compare *different mask selection techniques'* success under full parameter fine-tuning, only that full fine-tuning is much more successful than LoRA (Table 6).  If I have missed the relevant sections or results, I'll be pleased to learn about them.  I believe your result is different, though, and you deserve credit for showing it, as it indicates that channels cannot learn to compensate for neighboring channels' losing too much information, at least by simple methods, and it is important to allocate the sparsity correctly prior to tuning the model.

---

> > > > > > > ### Author Response · Authors · 2025-08-08
> > > > > > >
> > > > > > > Thank you so much for the recognition of our responses. We are glad to see that you have raised your recommendation.
> > > > > > >
> > > > > > > We sincerely appreciate your insight for our experiment that channels cannot learn to compensate for neighboring channels that lose too much information. Considering the prior works mentioned above, we thank you for the clarification that LLM-Pruner uses LoRA fine-tuning, yet we think its experiments demonstrate that a more flexible pruning pattern helps to reduce the performance degradation after the recovery process, which helps us to gain insights into the inherent sparsity allocation of LLMs. For Table 6 in Wanda, we think the unstructured sparsity offers more flexibility than the N:M sparsity, and its results show that full parameter fine-tuning is not able to fully compensate the performance gap between them.
> > > > > > >
> > > > > > > Thanks again for your insightful comments, which indeed help to improve our work. We are happy to answer further questions if you have any in the future.

---

> > > > > > > > ### Comment · Reviewer_wteP · 2025-08-08
> > > > > > > >
> > > > > > > > > For Table 6 in Wanda, we think the unstructured sparsity offers more flexibility than the N:M sparsity, and its results show that full parameter fine-tuning is not able to fully compensate the performance gap between them.
> > > > > > > >
> > > > > > > > I believe this may be the key - Lua-LLM allows for more flexibility compared to Wanda, in the same way that unstructured sparsity is more flexible than N:M sparsity.  In this way, it is different than simply choosing a better mask given the same constraints, which often makes no meaningful difference after substantial full parameter fine-tuning.
> > > > > > > >
> > > > > > > > Thank you for your remarks!

---

### Official Review · Reviewer_iAjF · 2025-07-02

**Clarity:** 3
**Significance:** 3
**Originality:** 2
**Rating:** 4
**Confidence:** 3

**Summary:**

The paper introduces Lua-LLM, a gradient-based global pruning framework for LLMs that learns unstructured sparsity allocation. Existing layer-wise pruning methods for LLMs often lead to suboptimal solutions by focusing on local errors, so Lua-LLM aims to address this by decomposing global pruning into row-wise subproblems. It uses a soft Top-K operator with a sigmoid function to approximate binary mask selection, enabling end-to-end optimization of sparsity thresholds.

**Questions:**

1. How to address the train-test mismatch between soft masks and hard binary pruning?

2. Please add ablation studies on different pruning granularities.

3. How to address the output-side weight importance in your method?

**Ethical Concerns:**

["NO or VERY MINOR ethics concerns only"]

**Final Justification:**

Thank you for the authors to address the concerns in the first round of review, I would like to raise the scores towards boarderline accept.

**Limitations:**

Yes (Have the authors adequately addressed the limitations and potential negative societal impact of their work? If so, simply leave “yes”)

**Quality:**

2

**Strengths And Weaknesses:**

**Strengths**

- The method consistently outperforms state-of-the-art techniques (e.g., ATP, DSA, OWL), particularly at extreme sparsity levels (e.g., 80%). Reductions in perplexity and improvements in zero-shot accuracy demonstrate its effectiveness in preserving model capability during compression.

- Lua-LLM learns sparsity allocation in as little as 1 hour for LLaMA-7B on 2× A100 GPUs, significantly faster than evolutionary methods like DSA (12+ hours).

**Weakness**

- Lua-LLM uses a sigmoid function to approximate Top-K selection during training (soft masks), but inference requires hard binary masks. This creates a fundamental conflict: training optimizes probabilistic weight retention, while inference enforces deterministic pruning. This inconsistency is not adequately addressed.

- The paper emphasizes row-wise pruning as a key advantage but fails to compare it against other granularities (e.g., column-wise, block-wise) in ablation experiments. (I understand this is something oberved in Wanda, but you should verify it in your experiments, since it is a key design).

- The authors acknowledge that output dimensions contain critical weights and notes that Wanda’s input-driven metric risks removing sensitive output-side weights. However, Lua-LLM retains Wanda’s design without incorporating explicit output-side importance signals, failing to address this limitation in its own sparsity allocation strategy.

---

> ### Author Rebuttal · Authors · 2025-07-31
>
> We are glad that the reviewer found that our work effectively preserves the model capability and achieves superior search efficiency. We thank the reviewer for the constructive feedback and appreciate the opportunity to address the questions you have raised.
>
> ---
>
> > Q1: The inconsistency that training with soft masks while inference with binary hard masks, is not adequately addressed.
>
> > Q4: How to address the train-test mismatch between soft masks and hard binary pruning?
>
> **A1**: Our Lua-LLM employs the **straight-through-estimator (STE) technique** [1], enabling gradient computation via a soft approximation during back-propagation when processing non-differentiable functions. As discussed in Section 2, STE techniques have also been adopted in other learning-based pruning methods such as MaskLLM [2] and DISP-LLM [3]. In this paper, we further propose a novel approximation method based on the sigmoid function, which provides an effective approximation for the Top-K mask selection function. For the sigmoid-based soft mask in Equation (7), we set the value of parameter *λ* to the number of elements per-row, which is large enough to provide a precise approximation. **During the training stage**, we use the sigmoid-based soft mask in Equation (7) and search for sparsity allocation with **learnable threshold parameters**. After the training process, we **store these threshold parameters and use the hard mask selection function in Equation (6) to generate hard masks with pruning thresholds for the inference stage**. We will incorporate the detailed interpretation of the straight-through-estimator technique into Section 4.2 of our revised manuscript.
>
> > Q2: The paper emphasizes row-wise pruning as a key advantage but fails to compare it against other granularities.
>
> > Q5: Please add ablation studies on different pruning granularities.
>
> **A2**: We conduct **ablation experiments under different** **pruning** **granularities**, i.e. different comparison group selection strategies. The results (**See Table below**) demonstrate that row-wise comparison remains the optimal choice for the adaptive sparsity allocation scenario, aligning with Wanda's observations for uniform pruning strategies, which is discussed in Section 3.2 of our manuscript.
>
> *Table 1: Perplexity of Lua-LLM with LLaMA2-7B on WikiText-2 under different pruning granularities.*
>
> | Granularity | 50%      | 60%      | 70%       | 80%       |
> | ----------- | -------- | -------- | --------- | --------- |
> | Row-wise    | **6.85** | **8.54** | **12.92** | **30.27** |
> | Column-wise | 7.16     | 9.42     | 14.17     | 51.76     |
> | Layer-wise  | 7.32     | 14.40    | 210.48    | 28599.64  |
>
> > Q3: Lua-LLM retains Wanda’s design without incorporating explicit output-side importance signals, failing to address this limitation in its own sparsity allocation strategy.
>
> > Q6: How to address the output-side weight importance in your method?
>
> **A3**: We thank the reviewer for raising this meaningful discussion for output-side importance signals. We would like to clarify that **Lua-LLM do not manually allocate sparsity based on** **heuristic** **statistics (e.g., output-side importance signals), in order to avoid suboptimal** **pruning** **results**. Instead, we adopt a NAS-based pruning strategy, which **automatically capture the output-side sensitivity to weight pruning, using gradient-based** **optimization** **techniques**. Our method decomposes the mask selection process into two components: intra-row and inter-row processing. (a) During **intra-row weight comparison**, all weights are within the same output channel. Consequently, calculating their importance metrics does not require considering output-side weight importance, and we can directly use Wanda's metric in Equation (2) to capture the weight importance with input-side signals. (b) For **inter-row sparsity allocation**, we do not manually allocate sparsity based on heuristic statistics, but with a NAS-based pruning strategy. This approach allows us to search for the optimal solution and control the overall sparsity via a regularization loss, eliminating the need for manual determination of per-row sparsity based on importance scores.
>
> ---
>
> Thanks for the attentive reading of the manuscript and constructive feedback. We hope our response addresses all the concerns and that the reviewer will consider raising the rating accordingly. We are glad to discuss further questions and suggestions.
>
> ---
>
> [1] Bengio, Yoshua, Nicholas Léonard, and Aaron Courville. "Estimating or propagating gradients through stochastic neurons for conditional computation." *arXiv preprint arXiv:1308.3432* (2013).
>
> [2] Fang, Gongfan, et al. "Maskllm: Learnable semi-structured sparsity for large language models." *Advances in Neural Information Processing Systems* 37 (2024): 7736-7758.
>
> [3] Gao, Shangqian, et al. "Disp-llm: Dimension-independent structural pruning for large language models." *Advances in Neural Information Processing Systems* 37 (2024): 72219-72244.

---

### Official Review · Reviewer_Gajc · 2025-07-02

**Clarity:** 3
**Significance:** 3
**Originality:** 3
**Rating:** 5
**Confidence:** 4

**Summary:**

Lua-LLM is a novel pruning framework for LLMs. The method learns an optimal unstructured sparsity pattern on a layerwise and within-layer basis. Global pruning is prohibitively complex, and existing layerwise pruning metrics lead to suboptimal parameter allocations. Lua-LLM outperforms other LLM pruning methods, particularly at high sparsity levels, and can lead to inference speedups due to integration with a preexisting framework.

**Questions:**

1. What are the speedups of other pruning methods (with or without the SpInfer framework)?
2. Could you compare the complexity of performing pruning on Lua-LLM vs other methods? I see section A addresses this, but some numbers would be nice

**Ethical Concerns:**

["NO or VERY MINOR ethics concerns only"]

**Final Justification:**

I was satisfied with the authors’ response and think this is a solid paper.

**Limitations:**

Yes

**Quality:**

3

**Strengths And Weaknesses:**

Overall, I think this is a solid paper with an interesting method and sufficient experimentation to prove the efficacy of the method.

Strengths:
- The paper introduces a technically novel method and clearly explains the process both intuitively and in rigorous detail.
- The proposed method outperforms existing baselines.
- The method leads to both lower parameter count and improved inference speed.
- Generally clear writing, good use of mathematical formalism, nice figures and plots.

Weaknesses:
- (See questions) The method seems to be computationally more expensive than some of its comparison points. This is fine but a detailed analysis of the complexity would be appreciated.
- An algorithm could help with clarity. The description of the method is a bit spread out, and I think a concise algorithm, even if it is in the appendix, could help provide a clear pointer to the steps in the method.

---

> ### Author Rebuttal · Authors · 2025-07-31
>
> We are glad that the reviewer found that our work is technically novel and clearly written. We thank the reviewer for the constructive feedback and appreciate the opportunity to address the questions you have raised.
>
> ---
>
> > Q1: Detailed analysis of the complexity.
>
> > Q4: Could you compare the complexity of performing pruning on Lua-LLM vs other methods?
>
> **A1**: Following the suggestions, we compare the **pruning time costs of Wanda, OWL, DSA, and Lua-LLM for LLaMA-7B model**  (**See Table below**). We agree with the reviewer that NAS-based pruning methods (DSA, Lua-LLM) are computationally more expensive than proxy-based methods (Wanda, OWL), and we would like to consider this as a meaningful trade-off for the significant performance improvements. Besides, we would like to clarify that our Lua-LLM (1 hour) is significantly faster than evolutionary methods like DSA (12+ hours), since we leverage a soft Top-K selection function to enable an efficient row-wise mask learning process.
>
> *Table 1:* *Pruning* *time comparisons for different pruning methods.*
>
> | Method | Wanda | OWL   | DSA  | Lua-LLM |
> | ------ | ----- | ----- | ---- | ------- |
> | Time   | 4 min | 6 min | 12 h | 1 h     |
>
> > Q2: An algorithm could help with clarity.
>
> **A2**: We thank the reviewer for the meaningful suggestion and we will incorporate an algorithm into our revised manuscript. Furthermore, we summarize the algorithm of our Lua-LLM into three stages as follows:
>
> (a) **Initialization**. We compute the importance scores for weight parameters with Equation (2), and uniformly initialize the row-wise threshold parameter t as the target sparsity p.
>
> (b) **Mask Learning**. We set the training loss with Equation (10). In the training process, we generate row-wise pruning masks with Equation (7) for forward-propagation, and update the learnable threshold parameters with back-propagation. After the training process, we save the learned threshold parameters.
>
> (c) **Pruning**. We use the threshold parameters and mask selection function in Equation (6) to prune the original model.
>
> > Q3: What are the speedups of other pruning methods (with or without the SpInfer framework)?
>
> **A3**: We evaluate the **inference speedups of Wanda and Lua-LLM for OPT-6.7B model** at 50%-70% sparsity levels. The experiment (**See Table below**) shows that the speedups of different pruning methods are quite the same for each sparsity level, which is compatible with the parameter counts. The results demonstrate that although adaptive unstructured pruning methods introduce a more irregular pattern for the sparse weight matrices, we can achieve meaningful speedup for the overall model with multiple performance optimization techniques employed in SpMM kernels like SpInfer, such as efficient sparse format and fine-grained execution pipeline.
>
> *Table 2: Inference throughput (token/s) speedups for different* *pruning* *methods.*
>
> | Method  | Sparsity             | Dense | 50%   | 60%    | 70%    |
> | ------- | -------------------- | ----- | ----- | ------ | ------ |
> | Wanda   | Throughput (token/s) | 696.6 | 844.9 | 1013.5 | 1200.6 |
> |         | Speedup              | -     | 1.21x | 1.45x  | 1.72x  |
> | Lua-LLM | Throughput (token/s) | 696.6 | 842.8 | 1012.7 | 1202.3 |
> |         | Speedup              | -     | 1.21x | 1.45x  | 1.73x  |
>
> ---
>
> Thanks for the attentive reading of the manuscript and constructive feedback. We hope our response addresses all the concerns and that the reviewer will consider raising the rating accordingly. We are glad to discuss further questions and suggestions.

---

> > ### Comment · Reviewer_Gajc · 2025-08-06
> >
> > Thank you to the authors for addressing the points in my review. I appreciate the additional results and explanations. I generally agree with the other reviewers and it does not change my evaluation of the paper. In my opinion, the authors sufficiently addressed the concerns raised by reviewer wteP. I would like to maintain my original score.

---

> > > ### Author Response · Authors · 2025-08-09
> > >
> > > We would like to express our sincere gratitude for your response and support on our paper. We are glad to see that our responses have addressed your points. Thank you for the thoughtful and constructive feedbacks, which help to improve our paper a lot.

---

### Decision · Program_Chairs · 2025-09-17

**Decision:**

Accept (poster)

**Comment:**

The authors propose a learning-based global pruning framework for optimal unstructured sparsity allocation in LLMs. The method leverages a soft Top-K operator to approximate the importance-based mask selection mechanism. The experimental results on LLaMA and OPT families demonstrate the effectiveness of the proposed approach.

The paper is technically sound and novel, and it clearly motivates and explains the methodology. The claims of the authors are supported by empirical evidence, showing that the method outperforms baselines (eg, in terms of number of parameters and inference speed). The authors also provided an extensive rebuttal and addressed all the concerns of the reviewers. For instance, the authors clarified the theoretical foundation of the paper and provided additional convincing empirical evidence. All reviewers voted for acceptance.